# Zebrafish imaging reveals TP53 mutation switching oncogene-induced senescence from suppressor to driver in primary tumorigenesis

Yukinari Haraoka[1], Yuki Akieda [1], Yuri Nagai[1], Chihiro Mogi[2] & Tohru Ishitani [1,2,3✉]

Most tumours are thought to arise through oncogenic cell generation followed by additional mutations. How a new oncogenic cell primes tumorigenesis by acquiring additional mutations remains unclear. We show that an additional TP53 mutation stimulates primary tumorigenesis by switching oncogene-induced senescence from a tumour suppressor to a driver. Zebrafish imaging reveals that a newly emerged oncogenic cell with the $Ras^{G12V}$ mutation becomes senescent and is eliminated from the epithelia, which is prevented by adding a TP53 gain-of-function mutation ($TP53^{R175H}$) into $Ras^{G12V}$ cells. Surviving $Ras^{G12V}$-$TP53^{R175H}$ double-mutant cells senesce and secrete senescence-associated secretory phenotype (SASP)-related inflammatory molecules that convert neighbouring normal cells into SASP factor-secreting senescent cells, generating a heterogeneous tumour-like cell mass. We identify oncogenic cell behaviours that may control the initial human tumorigenesis step. Ras and TP53 mutations and cellular senescence are frequently detected in human tumours; similar switching may occur during the initial step of human tumorigenesis.

---

[1] Department of Homeostatic Regulation, Division of Cellular and Molecular Biology, Research Institute for Microbial Diseases, Osaka University, Suita, Osaka 565-0871, Japan. [2] Institute for Molecular & Cellular Regulation, Gunma University, Gunma 371-8512, Japan. [3] Center for Infectious Disease Education and Research (CiDER), Osaka University, Suita, Osaka 565-0871, Japan. ✉email: ishitani@biken.osaka-u.ac.jp

Tumour cells harbour multiple mutations. Decades of human clinical sample analyses and animal model studies have revealed that mutations in oncogenes and tumour-suppressor genes synergistically promote tumorigenesis[1,2]. The biochemical interactions between these multiple mutations in cancer cell proliferation, survival, and migration have also been widely investigated[3–7]. Based on these studies, it is thought that most cancers originate from a sporadically emerged single oncogenic cell with a mutation in oncogenes or tumour-suppressor genes[8–10] and that mutation accumulation facilitates tumorigenesis[11–13]. However, the processes by which a single transformed cell primes neoplastic development and the effects of additional mutations on these processes remain unclear. Moreover, although a newly emerged oncogenic cell in healthy tissue is surrounded by normal cells, the interaction between an oncogenic cell and neighbouring normal cells in the early stages of tumorigenesis has not been extensively studied.

Recent studies suggested that cellular senescence, the state of ultimate and irreversible cell cycle arrest, plays both tumour-suppressive and tumour-promoting roles depending on the cellular context. Cellular senescence is triggered by various stresses, including chromosomal instability, telomere shortening, oxidative stress, and DNA damage[14–17]. Abnormal oncogenic activation, including hyperactivation of Ras signalling, can also induce cellular senescence to exert a tumour-suppressive role by inducing cell cycle arrest. In fact, senescence-associated cell cycle arrest has been detected in human premalignant tumours, and in vivo model studies using Ras signalling-activated or PTEN-deficient mice demonstrated that cellular senescence negatively regulates tumour initiation and progression[14,18–20]. However, senescent cells can also mediate tumorigenic effects by secreting a variety of inflammatory cytokines and growth factors, referred to as the senescence-associated secretory phenotype (SASP)[21]. SASP secretion leads to various effects, including a pathological increase in proliferation of neighbouring premalignant and malignant cells[22–24]. However, the roles of SASP in primary tumorigenesis, particularly in newly emerged oncogenic cells in living tissues, are unknown. In addition, how oncogenic cells decide to manipulate cellular senescence for tumour-suppressive or tumorigenic effects is unclear.

In this study, using imaging analyses, we show that a newly emerged Ras[G12V] oncogenic cell is eliminated from living tissue through cellular senescence. An additional TP53 mutation alters the effects of cellular senescence in Ras[G12V] cell from tumour-suppressive to SASP-mediated tumorigenic, and SASP secretion from the Ras-TP53 double mutant cell converts its neighbouring normal cells into senescent cells, thereby priming neoplastic development.

## Results

**Mosaically introduced Ras[G12V] cells are eliminated**. To understand the behaviour of a newly generated oncogenic cell, we used zebrafish larval skin as a model of human epithelial tissue. We previously established a method for oncogenic cell visualisation in zebrafish skin[25,26], in which fluorescent protein (e.g., mKO2, mCherry, or GFP)-expressing oncogenic cells were artificially introduced into the larval skin of the *keratin4* (*krt4*)-*gal4* zebrafish line[27] by injecting UAS promoter-driven expression plasmids (Fig. 1a). Low-dose injection of the UAS-driven plasmids induced a mosaic distribution of fluorescent cells, each of which was surrounded by normal cells in the larval skin (Supplementary Fig. 1a). Ras mutations, observed in ~30% of human cancers, are considered to occur in the early stages of tumorigenesis[28,29]. Previous studies demonstrated that forced activation of oncogenic Ras signalling covering a wide area can

induce excessive proliferation and dysplasia in the epithelial tissues of model animals, including mice, zebrafish, and flies[30–32]. Interestingly, mosaically introduced Ras[G12V] cells gradually disappeared and therefore did not induce any lesions (Fig. 1b). Ras[G12V] cells increased their cytoplasmic and nuclear volumes and apically protruded (Fig. 1c, Supplementary Movie 1, 2), but the protruding Ras[G12V] cells were nonapoptotic (Supplementary Fig. 1b). Thus, the newly generated Ras[G12V] cells surrounded by normal cells were eliminated from the larval skin. This oncogenic cell elimination appears to occur in an immune cell-independent manner. Consistent with this idea, Ras[G12V] cells were also eliminated from zebrafish larvae that lacked myeloid lineage cells (macrophages, monocytes, and neutrophil) established by injection of morpholino antisense oligo (MO) against *spi1b* (also known as *pu.1*)[33] (Supplementary Fig. 1c, d). In addition, zebrafish larvae have not developed an adaptive immune system, as this system is not functionally mature until approximately 4 weeks post-fertilization[34]. Moreover, we found that the neighbouring cells accumulate F-actin at the contact sites with the protruding Ras[G12V] cells (Supplementary Fig. 1e), suggesting that neighbouring epithelial cells are involved in Ras[G12V] cell elimination. To determine whether similar elimination can occur in other types of oncogenic cells, we mosaically introduced cells overexpressing v-Src, Myc and TP53 hot-spot mutants (TP53[R175H], TP53[R248W] or TP53[R273H]), the abnormal activity of which can induce tumorigenesis[35–37]. Mosaic v-Src cells were eliminated as observed for Ras[G12V] cells (Supplementary Fig. 2a), whereas Myc-overexpressing cells and TP53 mutants-expressing cells remained (Supplementary Fig. 2b–e), suggesting that this elimination is triggered by a specific type of oncogenic activation.

**Ras[G12V] cells undergo cellular senescence**. Cytoplasmic and nuclear swelling is a common phenotype in senescent cells[38,39]. In addition, activation of oncogenic Ras signalling can induce cellular senescence[38]. Therefore, we tested whether cellular senescence occurred in swollen Ras[G12V] cells in the zebrafish epithelia. The senescence phenotype is often characterised by several unique features: (i) activation of a chronic DNA damage response, such as phosphorylation of the histone H2AX (γH2AX), (ii) senescence-associated β-galactosidase (SA-β-gal) accumulation, (iii) epigenetic alternation such as H3K9 tri-methylation (H3K9me3), and (iv) irreversible cell-cycle arrest and its associated expression of various cyclin-dependent kinase inhibitors, including CDKN2A (p16[INK4a]), CDKN2B (p15[INK4b]), and CDKN1A (p21[CIP])[17,39]. In the zebrafish genome, *CDKN2A* and *CDKN2B* are integrated into only one gene, *cdkn2a/b*. Accumulation of γH2AX in the nucleus (Fig. 2a), SA-β-gal accumulation (Fig. 2b), H3K9me3 upregulation (Supplementary Fig. 3a), cell-cycle arrest (reduction of EdU incorporation) (Fig. 2c), and high *cdkn2a/b* expression (Fig. 2d) were detected in mosaically introduced Ras[G12V] cells, suggesting that the Ras[G12V] cells were senescent. Interestingly, high *cdkn2a/b* expression and nuclear size enlargement were not detected in zebrafish larval skin ubiquitously expressing Ras[G12V] (Supplementary Fig. 4a, b), indicating that Ras activation is not sufficient to induce cellular senescence in the skin. These results also suggest that neighbouring normal cells are required to induce senescence in mosaic Ras[G12V] cells.

**Senescence drives Ras[G12V] cell elimination**. To examine the involvement of cellular senescence in the Ras[G12V] cell elimination, we inhibited both CDKN2A/B- and TP53-mediated senescence-inducing pathways[17] by injecting MOs against *cdkn2a/b* and *tp53*. Injection of both MOs inhibited the swelling and elimination of mosaically introduced Ras[G12V] cells (Fig. 2e, f),

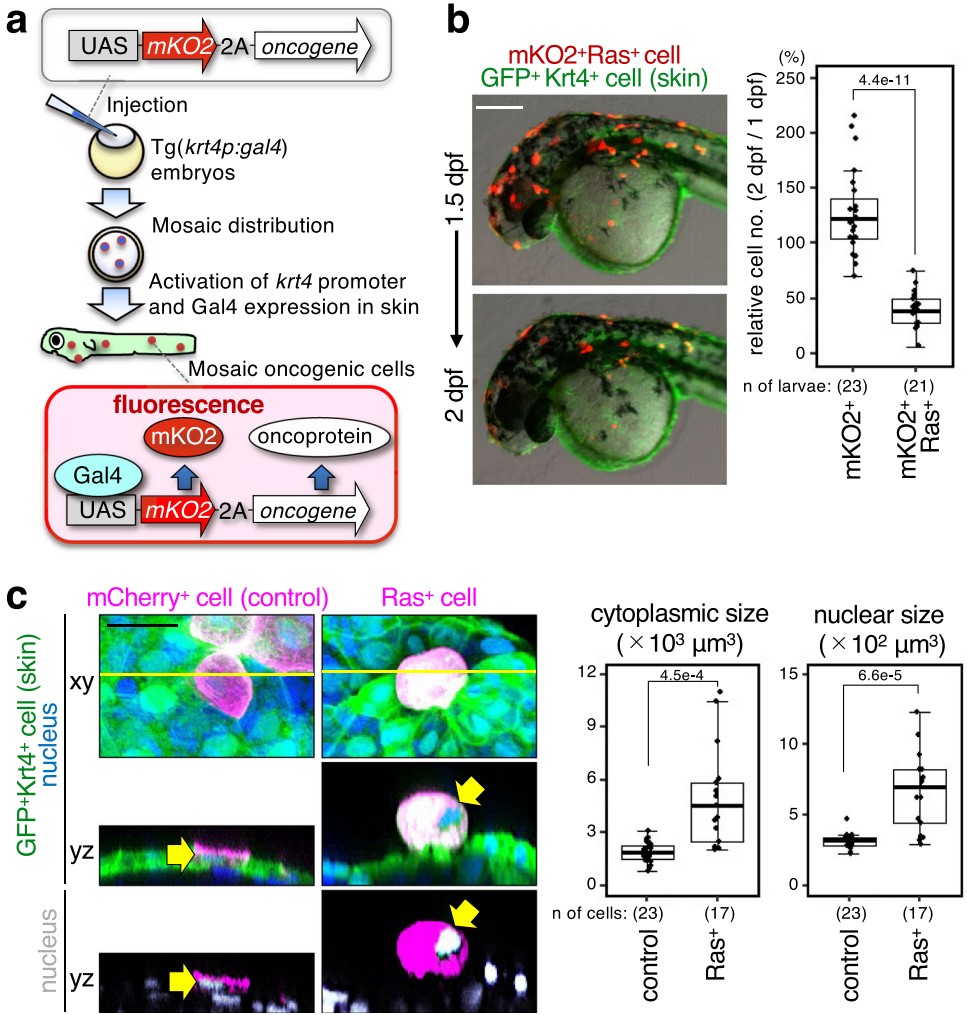

**Fig. 1 Newly generated Ras^{G12V} cell surrounded by normal cells is eliminated from zebrafish epithelia. a** Schematic illustration of experimental introduction of fluorescent oncogenic cells in zebrafish larval skin in a mosaic manner through the Gal4/UAS system. **b** Mosaic Ras^{G12V} cell disappears from larval skin. Representative images show the larvae with mosaically introduced cells expressing mKO2 with Ras^{G12V} (mKO2+Ras+ cells) (red). Skin cells were visualised with krt4p:gal4; UAS:EGFP (green). Scale bar: 200 μm. Bottom box plots of relative mKO2+ cell number of 2 days postfertilisation (dpf)/1 dpf ratio show first and third quartile, median is represented by a line, whiskers indicate the minimum and maximum, and outliers are shown as dots outside of the box. Each dot represents one larva. Unpaired two-tailed t-test was used. Note that oncogenic cells were introduced not only to the head region but also to the trunk and tail regions, and their disappearance occurred in all areas. **c** Cytoplasmic and nuclear swelling occurs in mosaic Ras^{G12V} cells. Representative confocal images show mosaic mCherry+ (control) or mCherry+Ras^{G12V} (Ras+) cells (magenta) and nucleus (upper; blue, lower; grey) in the larvae. Arrows indicates nuclei of mCherry+ cells. Scale bar: 20 μm. Box plots of estimated nuclear or cytoplasmic size (μm³) show first and third quartile, median is represented by a line, whiskers indicate the minimum and maximum, and outliers are shown as dots outside of the box. Each dot represents one cell. Unpaired two-tailed t-test was used. Source data are provided as a Source Data file.

suggesting that senescence-induced cell swelling generates a driving force for oncogenic cell elimination. We also examined the status of cell adhesion, as mosaically introduced Ras^{G12V} cells were apically delaminated. As expected, the membrane-localizing activity of E-cadherin, a major component of cellular adhesion[40], was decreased in the Ras^{G12V} cells, whereas the surviving Ras^{G12V} cells in larvae injected with cdkn2a/b and tp53 MOs retained membrane E-cadherin (Supplementary Fig. 5a), suggesting that Ras^{G12V}-induced cellular senescence reduces adhesion between Ras^{G12V} cells and neighbouring normal cells. Consistent with the above findings showing that ubiquitous expression of Ras^{G12V} cells was insufficient to induce cellular senescence in the skin, a reduction of membrane E-cadherin levels was not detected in the skin ubiquitously expressing Ras^{G12V} (Supplementary Fig. 5b), suggesting that neighbouring normal epithelial cells are required to induce senescence-mediated E-cadherin reduction and mosaic Ras^{G12V} cell elimination. These results indicate that cellular

senescence suppresses tumorigenesis through immune cell-independent oncogenic cell elimination.

**Ras^{G12V}-TP53^{R175H} double mutant cells form a tumour-like cell mass**. Although mutation accumulation is thought to facilitate tumorigenesis[11–13], the effects of additional mutations on the behaviour of a newly generated oncogenic cell remain unclear. Recently, we reported that a newly generated Wnt signalling-hyperactivated oncogenic cell is eliminated through apoptosis in zebrafish embryos, whereas loss of the tumour-suppressor gene smad4, which is associated with Wnt signalling hyperactivation in human cancers, blocked this elimination[25]. Therefore, we tested whether Ras^{G12V} cell behaviour is influenced by additional mutations, including gain or loss of TP53, gain of Myc, loss of Smad4, and loss of adenomatous polyposis coli, events which are known to interact genetically and/or biochemically with

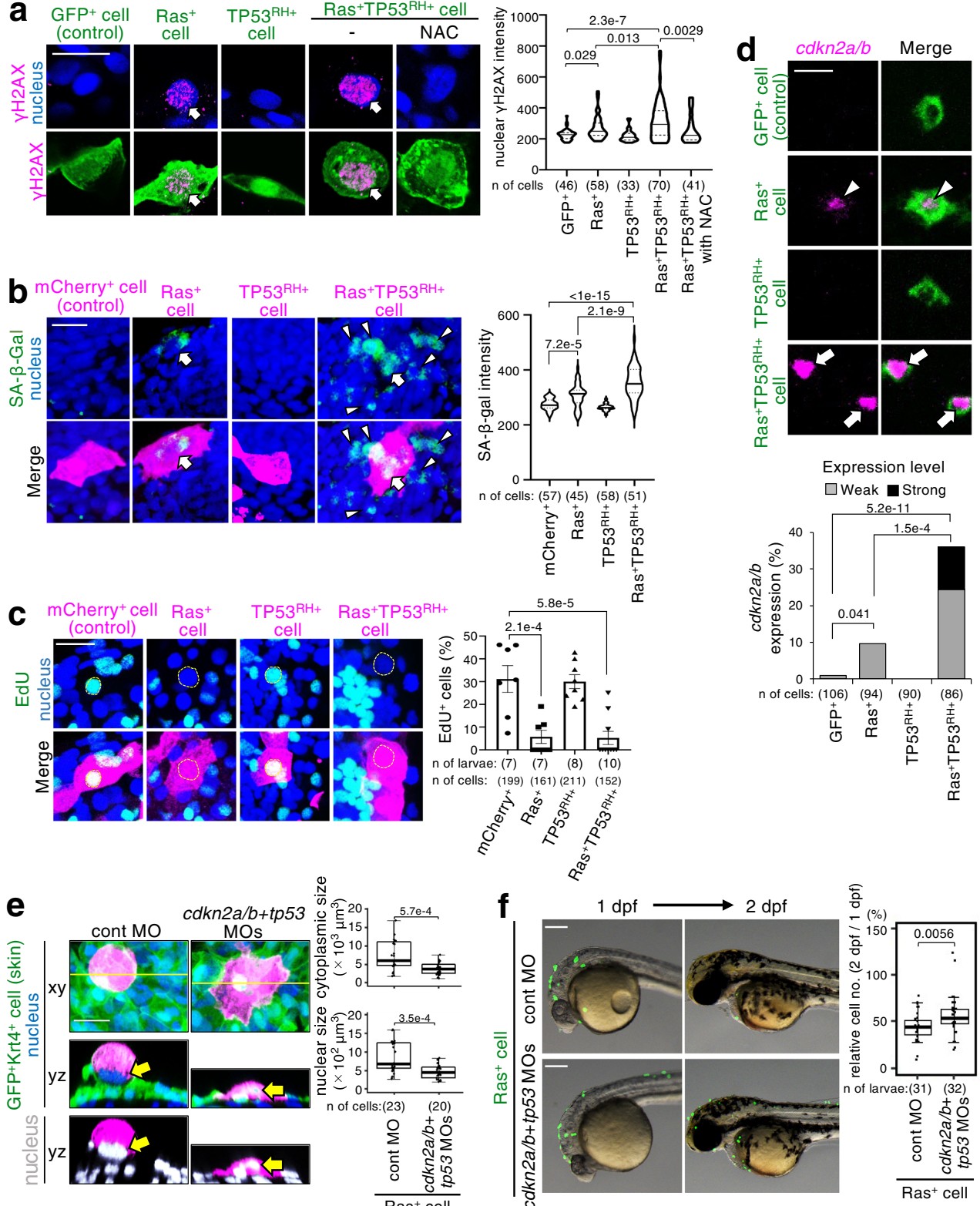

oncogenic Ras mutation in tumorigenesis[4–6,41,42]. Our analyses revealed that addition of TP53[R175H], TP53[R248W], or TP53[R273H], all of which are considered as TP53 gain-of-function hot-spot mutations[37], had the most significant effects on the Ras[G12V] cells. We mosaically introduced oncogenic cells co-expressing Ras[G12V] with each TP53 mutant into zebrafish larval skin (Supplementary

Fig. 6a). Although cells with Ras[G12V] alone were actively eliminated at 2 days postfertilisation (dpf), addition of TP53[R175H], TP53[R248W], or TP53[R273H] mutation to Ras[G12V] cells significantly increased their viability (Fig. 3a, Supplementary Fig. 6b). Consistent with these results, additional TP53[R175H] mutation blocked Ras[G12V] cell-induced F-actin accumulation in neighbouring cells

**Fig. 2 Newly generated oncogenic cells undergo cellular senescence. a–d** Representative confocal images show cells with GFP (**a, d**) or mCherry (**b, c**) alone (control), or with Ras$^{G12V}$, TP53$^{R175H}$, or Ras$^{G12V}$ and TP53$^{R175H}$ (GFP$^+$, mCherry$^+$, Ras$^+$, TP53$^{RH+}$, or Ras$^+$TP53$^{RH+}$ cells) (green in **a, d**, magenta in **b, c**). Scale bar: 20 μm. In **a**, arrows indicate γH2AX (magenta)-accumulated GFP$^+$ oncogenic cells. Violin plots of γH2AX intensities in GFP$^+$ cells show 75th, 50th (median), and 25th percentiles. In **b**, SA-β-gal was visualised by SPiDER-βGal (green). Arrowheads indicate SA-β-gal-expressing mCherry$^+$ oncogenic cells, and arrows indicate cells SA-β-gal-expressing neighbouring cells. Violin plots of SA-β-gal intensities in mCherry$^+$ cells show 75th, 50th (median), and 25th percentiles. In **c**, EdU$^+$ cells (green) were observed in mCherry$^+$ or TP53$^{RH+}$ cells, but not Ras$^+$ or Ras$^+$TP53$^{RH+}$ cells (magenta). Yellow broken lines: nuclei of mCherry$^+$ cells. Bar plots show EdU$^+$mCherry$^+$ cells (mean ± SEM). Each dot represents one larva. A two-tailed one-way ANOVA test with Sidak correction (**a, b, c**) was used. In **d**, fluorescent in situ hybridization visualised *cdkn2a/b* mRNA (magenta). Arrows or arrowheads indicates cells expressing *cdkn2a/b* weakly or strongly, respectively. Bottom graphs show percentages of GFP$^+$ cells with weak or strong *cdkn2a/b* expression. Fisher's exact test with Bonferroni correction was used. **e** Representative confocal images show cells expressing mKO2$^+$Ras$^{G12V}$ (Ras$^+$) (magenta) and nucleus (upper: blue, lower: grey). Skin cells visualised with *krt4p:gal4; UAS:EGFP* (green). Scale bar: 20 μm. Box plots of estimated nuclear or cytoplasmic size (μm$^3$) show first and third quartile, median is represented by a line, whiskers indicate the minimum and maximum. Each dot represents one cell. Unpaired two-tailed *t*-test was used. **f** Representative images show larvae with GFP$^+$Ras$^{G12V}$ (Ras$^+$) cells (green) with control MO or *cdkn2a/b* + *tp53* MOs. Scale bar: 200 μm. Box plots of relative Ras$^+$ cell number between 2/1 dpf ratio show first and third quartile, median is represented by a line, whiskers indicate the minimum and maximum, and outliers are shown as dots outside of the box. Each dot represents one larva. Unpaired two-tailed *t*-test was used. Source data are provided as a Source Data file.

(Supplementary Fig. 1e) and membrane E-cadherin reduction (Supplementary Fig. 5c). Interestingly, surviving Ras$^{G12V}$-TP53 double mutant cells formed heterogenous tumour-like cell masses containing a variety of cells, including surrounding skin cells lacking Ras$^{G12V}$-TP53 double mutations (Fig. 3b–d; Supplementary Fig. 6c; Supplementary Movie 3). E-cadherin levels in the cell mass were also heterogenous, and some double-mutant cells and their neighbours lost membrane E-cadherin (Supplementary Fig. 5d). In contrast, mosaically introduced cells with either Ras$^{G12V}$ or TP53$^{R175H}$ or with both Ras$^{G12V}$ and TP53 wild-type did not efficiently form the cell mass (Fig. 3d, e). Further, knockdown of endogenous *tp53* using *tp53* MO injection did not enhance Ras$^{G12V}$ cell-induced cell mass formation (Fig. 3d). Co-expression with TP53$^{DD}$, a TP53 dominant-negative mutant[43] that can inhibit the activity of TP53 wild-type (Supplementary Fig. 7a, b), in Ras$^{G12V}$ cells did not reinforce cell mass formation (Fig. 3d). Thus, gain-of-function, but not loss-of-function, of TP53 cooperates with the Ras$^{G12V}$ mutation to promote the evolution of a few oncogenic cells into a tumour-like cell mass. Notably, the introduction of cells co-expressing Ras$^{G12V}$ with a transcriptionally-inactive TP53$^{R175H-QSQS}$ mutant[44,45], did not induce cell mass formation (Fig. 3d), indicating that TP53$^{R175H}$ requires its transcriptional activity to facilitate double-mutant-derived cell mass formation.

**TP53$^{R175H}$ addition enhances Ras$^{G12V}$-induced senescence.** We next investigated how Ras$^{G12V}$ and TP53$^{R175H}$ mutations synergistically facilitate cell mass formation. Gain-of-function TP53 mutants, including TP53$^{R175H}$, acquire activities that elevate Ras$^{G12V}$-mediated signalling and gene expression[46], and TP53 positively regulates cellular senescence by controlling CDKN1A expression[47]. These previous reports suggest that TP53$^{R175H}$ affects Ras$^{G12V}$-mediated senescence. Consistent with this idea, co-expression of TP53$^{R175H}$ significantly enhanced Ras$^{G12V}$-induced expression of senescent makers, γH2AX, SA-β-gal, *cdkn2a/b*, and H3K9me3 (Fig. 2a, b, d, Supplementary Fig. 3a), and SASP factors, reactive oxygen species (ROS), interleukin (IL)-1β (*il1b*), IL-6 (*il6*), IL-6-related cytokine IL-11 (*il11b*), and IL-8/CXCL8 (*cxcl8a*)[21,39,48,49], were synergistically induced by co-expression of TP53$^{R175H}$ with Ras$^{G12V}$ (Fig. 4a, b, Supplementary Fig. 3b, c). These SASP factors were detected inside the Ras$^{G12V}$-TP53$^{R175H}$ double-mutant cells (Fig. 4a, white arrows in top panels and bottom left graph, and Fig. 4b, Supplementary Fig. 3b) and *il1b* expression was specifically activated in cell cycle-arrested senescent Ras$^{G12V}$-TP53$^{R175H}$ cells (Supplementary Fig. 3d). In contrast, the effects of TP53$^{R175H}$ mutation alone on the expression levels of these senescent makers and SASP factors were very minor (Figs. 2a, b, d, 4a, b, Supplementary Fig. 3a–c). In addition, depletion of endogenous TP53 proteins by *tp53* MO injection did not affect Ras$^{G12V}$-induced *cdkn2a/b* expression (Supplementary Fig. 7c). Co-expression of exogenous TP53 wild-type or TP53$^{DD}$ dominant-negative mutant with Ras$^{G12V}$ did not activate the expression of *il1b* and *il11b* (Supplementary Fig. 7d). These results suggest that gain-of-function, but not loss-of-function, of TP53 cooperates with the Ras$^{G12V}$ mutation to promote cellular senescence and SASP factor production.

**Double-mutant cells form tumour-like cell masses through SASP.** Although active cell proliferation should occur during cell mass formation, almost all Ras$^{G12V}$-TP53$^{R175H}$ double-mutant cells were EdU-negative non-proliferative cells (Fig. 2c) and EdU-positive double-mutant cells were not detected in the cell masses (Supplementary Fig. 8a). Interestingly, EdU-positive or phospho-histone H3 (pH3)-positive proliferating cells lacking Ras$^{G12V}$-TP53$^{R175H}$ mutations were detected in the double-mutant cell-surrounding area (Fig. 5a, Supplementary Fig. 8b). Surviving double-mutant cells in zebrafish skin express SASP factors, suggesting that SASP factors secreted from these cells stimulate the proliferation of the neighbouring cells and consequent cell mass formation. To test this, we knocked down the mitogenic SASP factor *il1b*[50–52] using MO blocking *il1b* mRNA translation (Supplementary Fig. 9a). As expected, *il1b* knockdown blocked the double-mutant cell-induced increase in pH3-positive proliferating cells in the surrounding area (Fig. 5a) and cell mass formation (Fig. 5b), suggesting that the double-mutant cell-derived Il1b stimulates cell proliferation in neighbouring normal cells, leading to cell mass formation. We also found that treatment with the ROS scavenger *N*-acetyl-L-cysteine (NAC) reduced ROS production in the double-mutant cells (Fig. 4a) and cell mass formation (Fig. 5c), suggesting that, in addition to Il1b, other SASP factor ROS are involved in cell mass formation. To further evaluate the contribution of Il1b and ROS, we artificially increased the levels of Il1b and ROS in mosaic Ras$^{G12V}$ cells. Interestingly, co-expression of exogenous Il1b in Ras$^{G12V}$ cells reduced F-actin accumulation in the neighbouring cells (Supplementary Fig. 9b) and induced cell mass formation in a small number of larvae (Fig. 5d), indicating that addition of exogenous Il1b partially mimicked the effects of TP53$^{R175H}$ mutation addition. Moreover, exposure to hydrogen peroxide (H$_2$O$_2$), an ROS that can be generated in senescent cells[17], further facilitated cell mass formation induced by cells co-expressing Ras$^{G12V}$ cells and exogenous Il1b (Fig. 5d). However, H$_2$O$_2$ treatment did not affect Ras$^{G12V}$ cell-induced cell mass formation in the absence of exogenous Il1b (Fig. 5d). These results suggest that the double mutation-induced

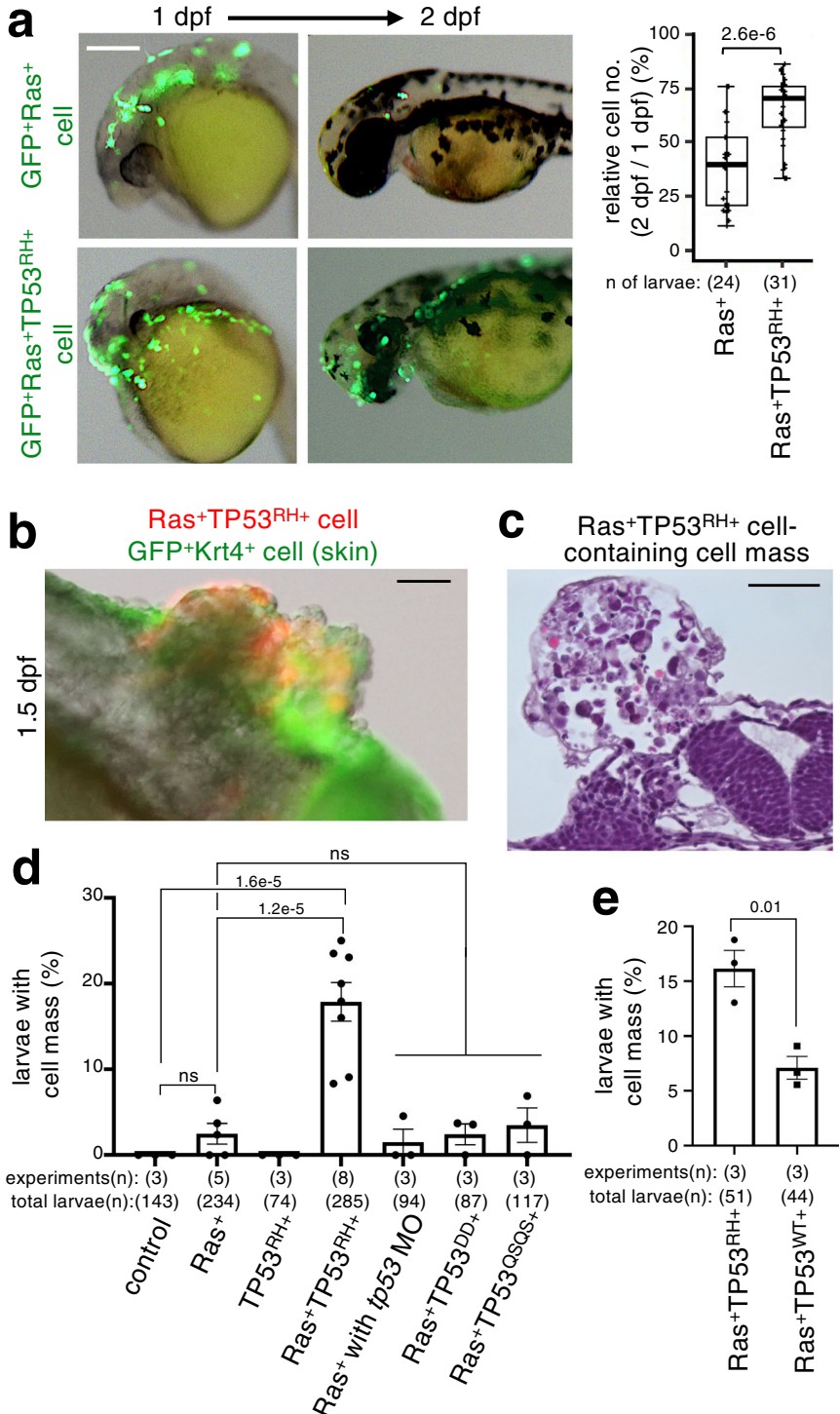

**Fig. 3 TP53^R175H mutation attenuates mosaic Ras^G12V cell elimination and stimulates heterogeneous cell mass formation. a** Mosaic Ras$^{G12V}$TP53$^{R175H}$ cells remain in larval skin. Representative images show larvae with mosaically introduced GFP$^+$Ras$^{G12V}$ (GFP$^+$Ras$^+$) or GFP$^+$Ras$^{G12V}$TP53$^{R175H}$ (GFP$^+$Ras$^+$TP53$^{RH+}$) cells (green). Scale bar: 200 μm. Box plots of relative GFP$^+$ cell number of 2 dpf/1 dpf ratio show first and third quartile, median is represented by a line, whiskers indicate the minimum and maximum. Each dot represents one larva. Unpaired two-tailed *t*-test was used. **b**–**e** TP53 gain of function, but not loss of function, stimulates heterogeneous cell mass formation. In **b**, representative images show cell mass in the trunk region of larvae with mosaically introduced mKO2$^+$Ras$^{G12V}$TP53$^{R175H}$ (Ras$^+$TP53$^{RH+}$) cells (red) at 1.5 dpf. Skin surface layer cells were visualised with *krt4p:gal4;* UAS:EGFP (green). In **c**, representative image of haematoxylin and eosin staining of cell mass in the trunk region of larvae with mosaically introduced Ras$^+$TP53$^+$ cells. Scale bar: 50 μm. Graph in **d** and **e** shows the percentage of 1.5 dpf embryos with cell mass (Means ± SEM). Each dot represents an independent experimental result. In **d**, A two-tailed one-way ANOVA test with Sidak correction was used. In **e**, Unpaired two-tailed *t*-test was used. Note that cell mass formation was induced not only in the head region but also in the trunk and tail regions. Source data are provided as a Source Data file.

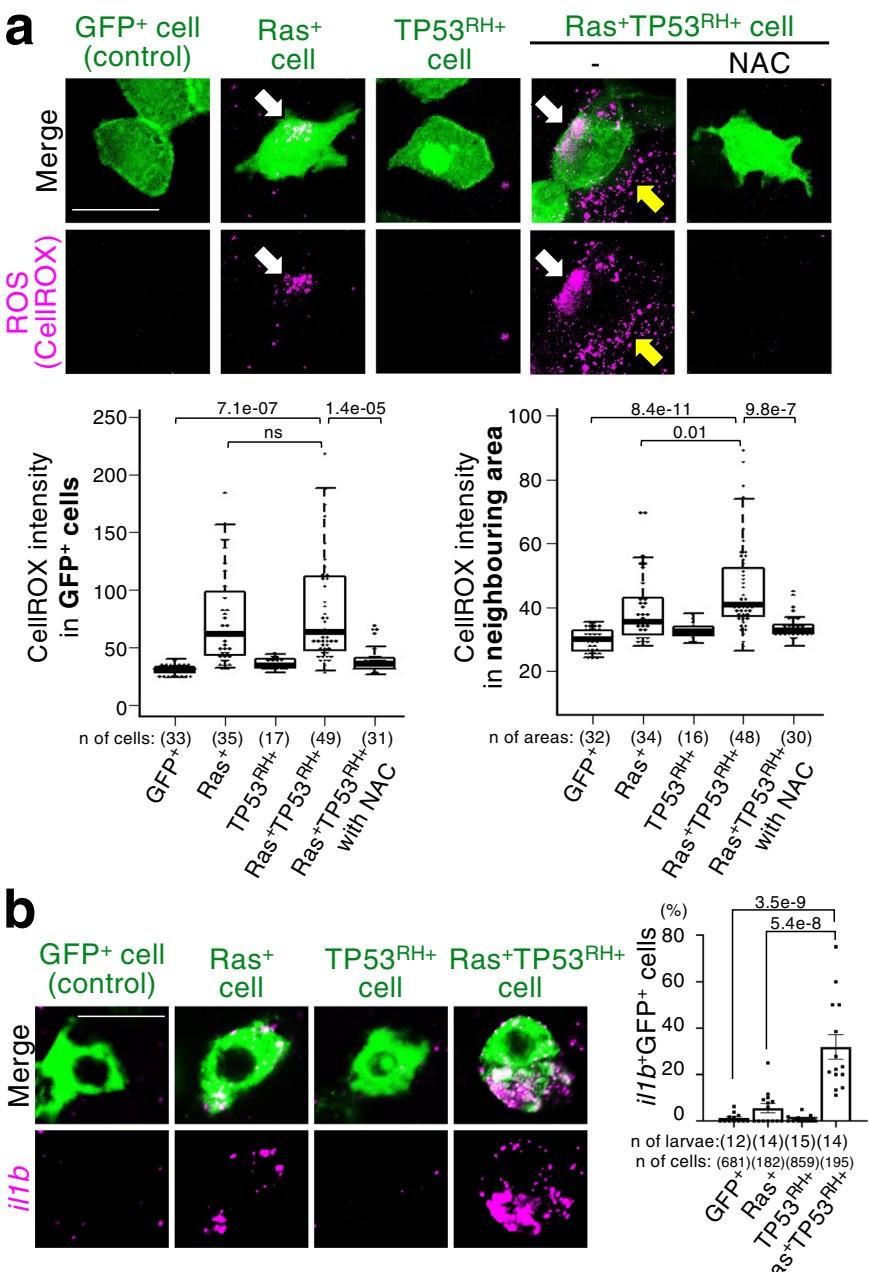

**Fig. 4 Additional TP53$^{R175H}$ mutation reinforces SASP. a** A SASP factor ROS is detected in mosaic Ras$^{G12V}$ and Ras$^{G12V}$TP53$^{R175H}$ cells and Ras$^{G12V}$TP53$^{R175H}$ cell neighbours. Representative confocal images show ROS-probe (CellRox DeepRed (magenta))-stained larvae with mosaically introduced cells expressing GFP alone (control) or with Ras$^{G12V}$, TP53$^{R175H}$, or both Ras$^{G12V}$ and TP53$^{R175H}$ (GFP+, Ras+, TP53$^{RH+}$, or Ras+TP53$^{RH+}$ cells) (green). White arrows indicate ROS in GFP+ oncogenic cells. Yellow arrows indicate ROS in the neighbouring area. NAC treatment reduces ROS. Scale bar: 20 μm. Box plots of CellROX intensities in GFP+ cells or neighbouring area (20 μm radius around GFP+ cells) show first and third quartile, median is represented by a line, whiskers indicate the minimum and maximum, and outliers are shown as dots outside of the box. Each dot represents one cell or one area. A two-tailed one-way ANOVA test with Sidak correction was used. **b** Ras$^{G12V}$ and TP53$^{R175H}$ synergistically promote the expression of SASP factors, interleukin-1β (*il1b*). Representative confocal images show fluorescence in situ hybridization of *il1b* mRNA (magenta) in larvae with mosaically introduced cells expressing GFP alone (control) or with Ras$^{G12V}$, TP53$^{R175H}$, or both Ras$^{G12V}$ and TP53$^{R175H}$ (GFP+, Ras+, TP53$^{RH+}$, or Ras+TP53$^{RH+}$ cells) (green). Scale bar: 20 μm. Bar plots show *il1b*+GFP+ cells (mean ± SEM). Each dot represents one larva. A two-tailed one-way ANOVA test with Sidak correction was used. Source data are provided as a Source Data file.

high-level production of Il1b and ROS in oncogenic cells enables cell mass formation in cooperation with neighbouring cells.

**Propagation of senescence promotes tumour-like cell mass formation.** Notably, the introduction of the double mutant cells upregulated ROS levels not only in themselves but also in

neighbouring cells, and introduction of cells with either Ras$^{G12V}$ or TP53$^{R175H}$ mutation did not induce such ROS diffusion (Fig. 4a, yellow arrows in top panels and bottom right graph). ROS were also detected in cells with and without double mutations inside the growing cell mass (Fig. 5e). Given that ROS are known both SASP factors[17,48] and activators of senescence through DNA double-strand

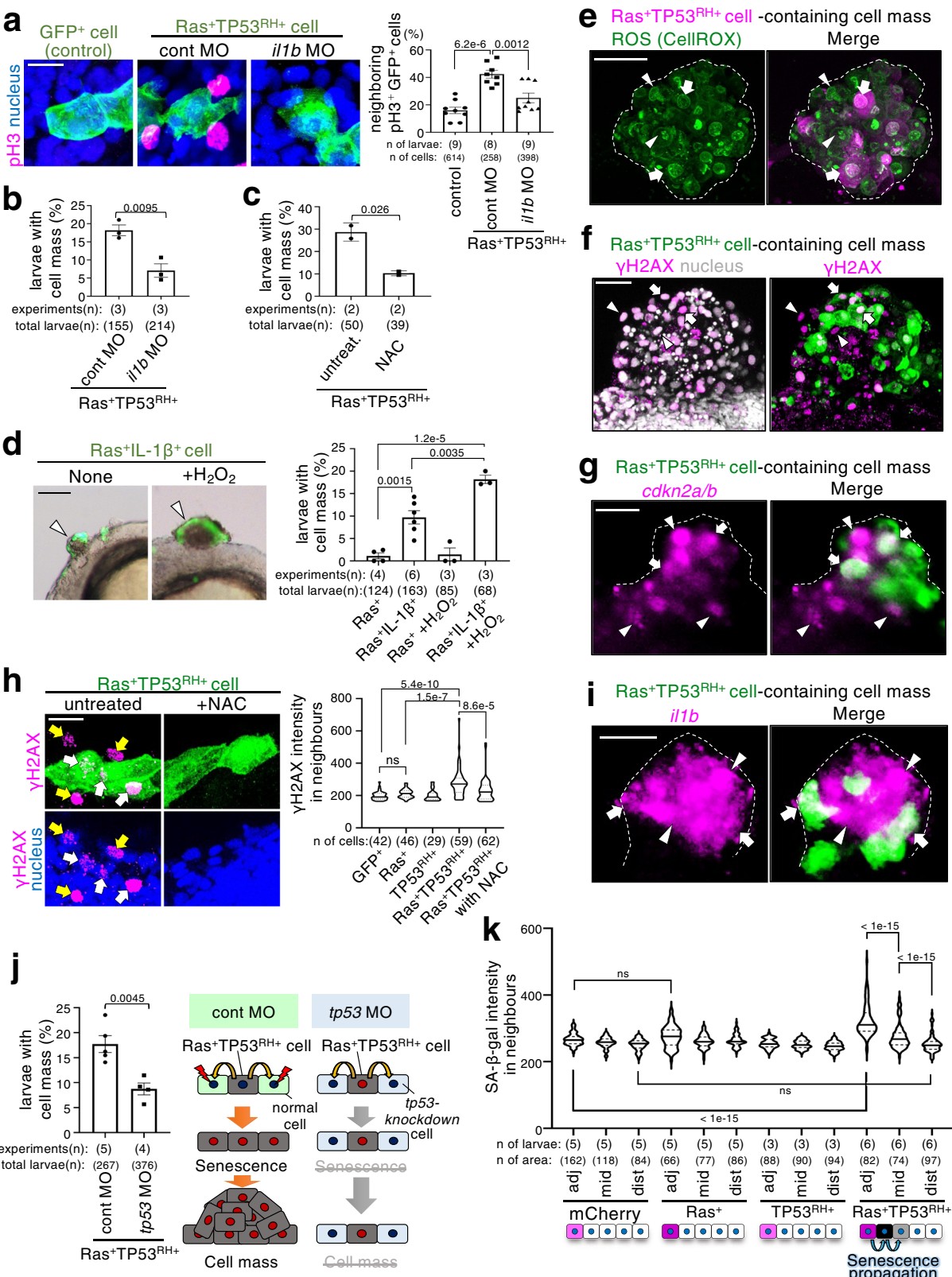

breaks induction[53], we hypothesised that diffused ROS induce senescence in neighbouring cells. Consistently, the growing cell mass contained γH2AX-, or *cdkn2a/b*-positive cells lacking Ras[G12V]-TP53[R175H] mutations (Fig. 5f, g, arrowheads). In addition, mosaically introduced Ras[G12V]-TP53[R175H] double-mutant cells, but not cells with either Ras[G12V] or TP53[R175H], induced SA-β-gal activity and

γH2AX in neighbouring cells (Fig. 2b, arrowheads, Fig. 5h). All γH2AX-positive neighbouring cells were pH3-negative non-proliferative cells (Supplementary Fig. 8b), and NAC treatment reduced the double-mutant-induced γH2AX in neighbours (Fig. 5h). These results suggest that senescent double-mutant cell-secreted ROS senesce neighbouring cells.

**Fig. 5 SASP-mediated senescence propagation is involved in Ras$^{G12V}$TP53$^{R175H}$ cell-driven cell mass formation. a** Representative confocal images show phospho-histoneH3 (pH3) (magenta) and nucleus (blue) in the larvae with GFP$^+$ (control) or Ras$^+$TP53$^+$ cells (green) with control or *il1b* MO. Scale bar: 20 μm. Graph shows the percentage of GFP$^+$ cells with pH3$^+$ neighbours (mean ± SEM). **b, c** Larvae with Ras$^+$TP53$^{RH+}$ cells with control or *il1b* MO (**b**) or untreated or treated with NAC (**c**). Graph shows the percentage of 1.5 dpf embryos with cell mass (mean ± SEM). **d** Representative images show cells co-expressing Ras$^{G12V}$ and zebrafish Il1b (Ras$^+$IL-1β$^+$cells) (green) with or without H$_2$O$_2$ treatment. Scale bar: 50 μm. Graph shows the percentage of 1.5 dpf embryos with cell mass (mean ± SEM). Each dot in graph (**a-d**) represents an independent experimental result. **e-g, i** Representative images show ROS (**e**; green), γH2AX (**f**; magenta), *cdkn2a/b* mRNA (**g**; magenta), or *il1b* mRNA (**i**; magenta) and nucleus (**f**; grey) in cell mass on the larvae with Ras$^+$TP53$^{RH+}$ cells (**e**; magenta, **f**, **g**, **i**; green). Dot lines surround cell mass. Scale bar: 50 μm. **h** Representative confocal images show γH2AX (magenta) and nucleus (blue) in larvae with Ras$^+$TP53$^{RH+}$ cells (green) untreated or treated with NAC. Scale bar: 20 μm. Violin plots of γH2AX intensities in neighbours (<20 μm radius from GFP$^+$, Ras$^+$, TP53$^{RH+}$, or Ras$^+$TP53$^{RH+}$ cells) show 75th, 50th (median), and 25th percentiles. **j** Schematic illustration (right) shows experimental inhibition of secondary senescence in neighbouring cells. Larvae with Ras$^+$TP53$^{RH+}$cells were injected with control or *tp53* MO. Graph show percentage of 1.5 dpf embryos with cell mass (means ± SEM). **k** Violin plots of SA-β-gal intensities in neighbouring cells around mCherry$^+$ cells show 75th, 50th (median), and 25th percentiles. adj, mid, and dist indicates SA-β-gal activity-positive adjacent cells (<20 μm radius from the double-mutant cells), second adjacent cells (20–40 μm radius), and distant cells (more than 40-μm radius). A two-tailed one-way ANOVA test with Tukey's correction (**a**) or Sidak correction (**d**, **h**, **k**) or unpaired two-tailed t-test (**b**, **c**, **j**) was used. Source data are provided as a Source Data file.

To further analyse neighbouring cell senescence, we generated cellular senescence reporter zebrafish Tg(*cdkn2a/b*-hs:Achilles) (Supplementary Fig. 10a), which express fast-maturing YFP variant protein, Achilles[54], in cells expressing endogenous *cdkn2a/b* (Supplementary Fig. 10b). As expected, the *cdkn2a/b*-hs:Achilles reporter was upregulated in mosaic Ras$^{G12V}$ single-mutant or Ras$^{G12V}$-TP53$^{R175H}$ double-mutant cells, and almost all of these cells were EdU-negative arrested (Supplementary Fig. 10c), suggesting that the *cdkn2a/b*-hs:Achilles reporter can be used to detect senescent cells. Time-lapse imaging of live Tg(*cdkn2a/b*-hs:Achilles) larvae captured the beginning of *cdkn2a/b* reporter expression in Ras$^{G12V}$-TP53$^{R175H}$ cells (Supplementary Fig. 10d, Supplementary Movie 4) and subsequent expression of the *cdkn2a/b* reporter in the neighbours (Supplementary Fig. 10e, Supplementary Movie 5). These observations are consistent with our model that the senescent double-mutant cells propagate senescence to the neighbours. We also confirmed that both *cdkn2a/b* reporter-positive senescent cells and EdU-positive proliferating cells were present in the Ras$^{G12V}$-TP53$^{R175H}$ cell neighbours, but these two populations were mutually exclusive (Supplementary Fig. 10f).

ROS and *il1b* and *il6* transcripts were detected in both Ras$^{G12V}$-TP53$^{R175H}$ double-mutant cells and cells lacking Ras$^{G12V}$-TP53$^{R175H}$ mutations inside the cell mass (Fig. 5e, i, Supplementary Fig. 11a), indicating that SASP factors are generated not only in the double-mutant cells but also in the senesced neighbours. To test this possibility, we prevented senescence in neighbouring normal cells by treatment with NAC to inhibit ROS, a senescence inducer secreted from double-mutant cells, or by MO-mediated knockdown of *tp53*, which is a mediator of ROS-induced cellular senescence[55]. NAC treatment reduced the expression of SASP factors, *il1b* and *il11b*, in larvae mosaically introduced with Ras$^{G12V}$-TP53$^{R175H}$ double-mutant cells (Supplementary Fig. 3e). In TP53-knockdown zebrafish skin, mosaic introduction of the double-mutant cells did not efficiently induce ROS production and *il1b* expression in the surrounding area (Supplementary Fig. 11b, c) and cell mass formation (Fig. 5j), suggesting that neighbouring cell senescence and its associated production of SASP factors are required for cell mass formation. These findings also indicate that ROS generated in senescent double-mutant cells induce senescence and ROS production in the neighbours, suggesting that a senesced neighbouring cell also induces senescence in its neighbours, which may propagate senescence. Consistent with this idea, mosaic introduction of the double-mutant cells induced strong SA-β-gal activity in adjacent cells (~20 μm radius from the double-mutant cells) and moderate SA-β-gal activity in second adjacent cells (20–40 μm radius from the double-mutant cells) (Fig. 5k, Supplementary Fig. 11d). In the

double-mutant cell-induced cell mass, the senescence markers γH2AX and *cdkn2a/b* were broadly expressed in not only the area immediately neighbouring the double-mutant cells but also 2–3 cells away from these cells (Fig. 5f, g). Thus, in addition to Il1b-mediated neighbouring cell proliferation, ROS-mediated senescence propagation may also contribute to cell mass formation.

**Mosaic Ras$^{G12V}$ cells form a cell mass in damaged skin.** Next, we tested the behaviour of oncogenic cells on unhealthy epithelia with abnormal TP53 activity or senesced epithelia. Mosaically introduced of Ras$^{G12V}$ mutation in larval skins ubiquitously expressing TP53$^{R175H}$ efficiently generated tumour-like cell masses (Fig. 6a), in which γH2AX, ROS, and *il1b* were detected in cells with and without Ras$^{G12V}$ mutation (Supplementary Fig. 12a–c); this cell mass formation was supressed by *il1b* MO or NAC treatment (Fig. 6b, c), suggesting that a newly generated Ras$^{G12V}$ mutation can prime tumorigenesis by inducing Il1b-mediated proliferation and ROS-mediated senescence in its neighbours in the epithelia with abnormal TP53 activity. To produce senesced epithelia, we treated zebrafish larvae with doxorubicin, which induces DNA damage and consequent TP53 activation and cellular senescence[56,57]. Doxorubicin treatment partially induced γH2AX-positive senescent cells in larval skin (Supplementary Fig. 12d), whereas it did not induce apoptotic cells (Supplementary Fig. 12e). Compared to mosaically introduced Ras$^{G12V}$ cells in normal skin, those in doxorubicin-treated skin relatively strongly expressed *cdkn2a/b*, *il1b*, *1l11b*, and ROS (Fig. 6d–g, Supplementary Fig. 12f) and efficiently induced tumour-like cell masses (Fig. 6h). Mosaic Ras$^{G12V}$ cell-induced cell mass formation in doxorubicin-treated skin was blocked by MO-mediated *tp53* knockdown (Fig. 6i). These results indicate that the mechanism of Ras$^{G12V}$ cell-induced cell mass formation in senesced epithelia is similar to that of double-mutant cell-induced cell mass formation in the healthy epithelia, and that senesced epithelia lose the potential to eliminate oncogenic cell and become prone to tumorigenesis initiation compared to healthy epithelia.

## Discussion
We demonstrated the potential mechanisms of tumour initiation and roles of cellular senescence in this process. In healthy zebrafish epithelial tissue, a newly emerged oncogenic Ras$^{G12V}$ cell becomes senescent, which stimulates cell swelling and loss of intercellular adhesion with neighbouring cells and consequent Ras$^{G12V}$ cell elimination (Fig. 7, left). However, the addition of a TP53 gain-of-function mutation (TP53$^{R175H}$) prevents Ras$^{G12V}$ cell elimination by restoring adhesion with neighbouring cells and reinforces Ras$^{G12V}$ cell senescence and consequent production of

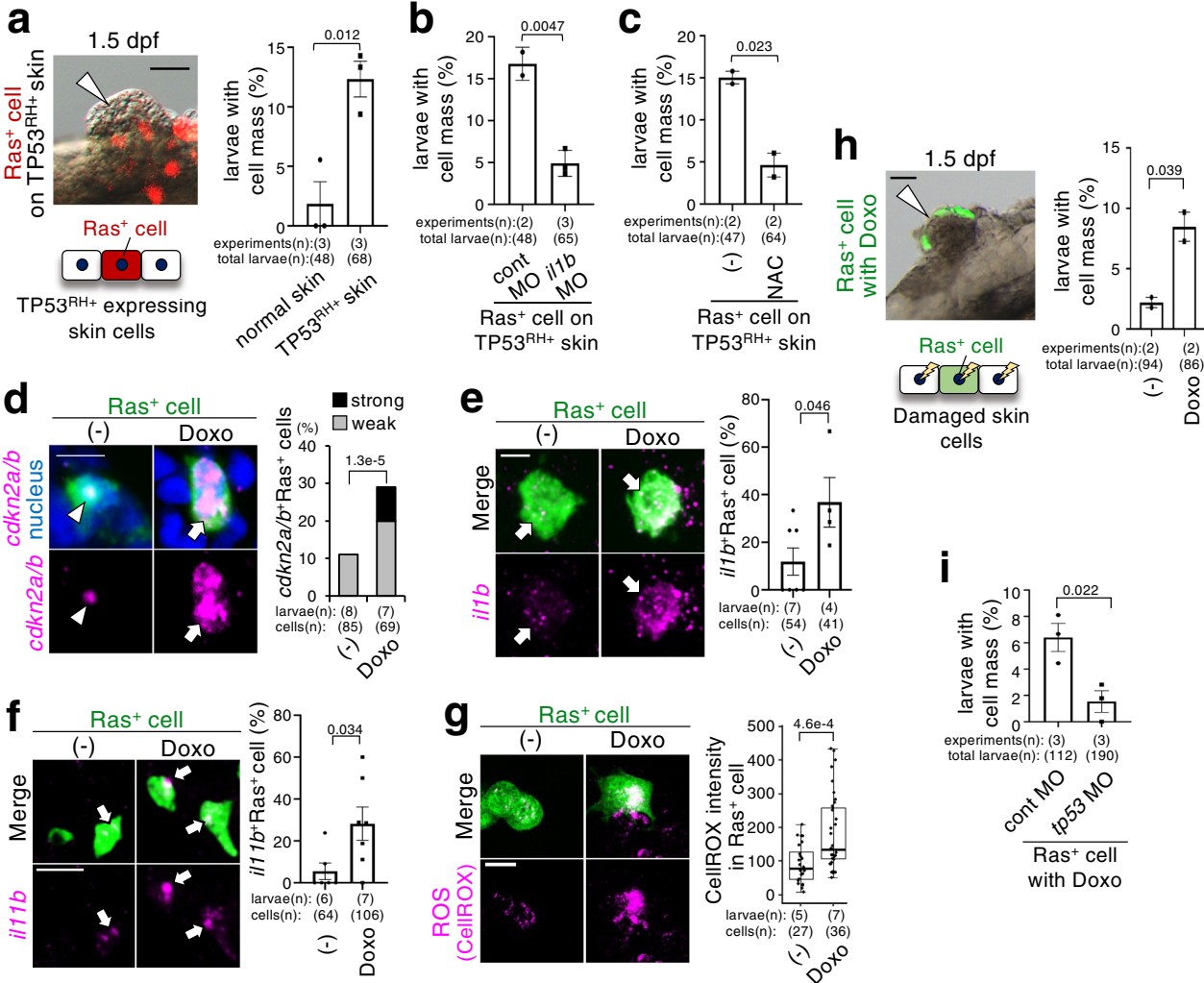

**Fig. 6 Mosaic Ras^G12V cells stimulate cell mass formation in damaged skin. a** Representative images show cell mass in Tg(krt4p: gal4; UAS:G43EGFP-TP53R175H) larval head region with mCherry⁺Ras⁺ (Ras⁺) cells (red). Graph shows percentage of embryos with cell mass (mean ± SEM). **b, c** Tg(krt4p: :gal4; UAS:G43EGFP-TP53R175H) larvae with Ras⁺ cells with control or il1b MO (**b**) or untreated or treated with NAC (**c**). Graph shows the percentage of 1.5 dpf embryos with cell mass (mean ± SEM). Each dot in graph (**a–c**) represents an independent experimental result. **d–g** Doxorubicin promotes senescence and SASP factor production in mosaic Ras⁺ cells. In **d**, representative images show fluorescence in situ hybridization for cdkn2a/b mRNA (magenta) in Ras⁺ cells (green) with or without doxorubicin treatment, and nucleus (blue). Arrow and arrowhead indicate cdkn2a/b⁺Ras⁺ cells weakly or strongly, respectively. Graph shows percentage of cdkn2a/b⁺Ras⁺ cells. In **e–f**, representative images show fluorescence in situ hybridization for il1b (**e**) or il11b mRNA (**f**) (magenta) in Ras⁺ cells (green) with or without doxorubicin treatment. Arrows indicate il1b⁺Ras⁺ or il11b⁺Ras^G12V cells. Graph shows percentage of il1b⁺Ras⁺ or il11b⁺Ras^G12V cells (mean ± SEM). Each dot represents one larva. In **g**, representative images show ROS-probe, CellROX (magenta), -stained larvae with Ras⁺ cells (green) with or without doxorubicin treatment. Box plots of CellROX relative intensities in Ras⁺ cells show first and third quartile, median is represented by a line, whiskers indicate the minimum and maximum. Each dot represents one cell. **h** Representative images show cell mass in the head region of doxorubicin-treated larvae with GFP⁺Ras⁺ (Ras⁺) cells (green). Right graph shows percentage of embryos with cell mass (mean ± SEM). **i** Doxorubicin-treated larvae with Ras⁺ cells with control or tp53 MO. Graph shows percentage of 1.5 dpf embryos with cell mass (mean ± SEM). Each dot in graph (**h, i**) represents an independent experimental result. Note that cell mass formation was induced not only in the head but also in the trunk and tail in TP53R175H-expressing and doxorubicin-treated skin. Scale bar: 50 (**a, h**) or 10 (**d–g**) μm. Unpaired two-tailed t-test (**a–c, e–i**) or Fisher's exact test with Benjamini–Hochberg correction (**d**) was used. Source data are provided as a Source Data file.

SASP factors, including IL-1β and ROS (Fig. 7, middle). Additional TP53 mutation-mediated survival of mutant cells would assist in the continuous high-level secretion of IL-1β and ROS to the neighbours. Neighbouring cells undergo proliferation or senescence in response to IL-1β or ROS, respectively. Secondary senesced cells also secrete IL-1β and ROS, which can propagate cellular senescence and stimulate aberrant cell proliferation nearby, resulting in formation of a heterogeneous tumour-like cell mass. Similar heterogeneous cell mass formation depending on IL-1β and ROS occurs in abnormal zebrafish skin with TP53R175H mutation or DNA damage (Fig. 7, right).

Previous studies revealed two tumour-suppressive mechanisms of cellular senescence. The first is induction of cyclin-dependent kinase inhibitor-mediated irreversible cell cycle arrest in oncogenic cells[14–17]. The second is stimulation of immune cell-mediated senescent cell elimination[58,59]. In this study, we found that immune cell-independent oncogenic cell elimination is a third antitumour mechanism of cellular senescence. Interestingly, similar elimination was triggered by activation of Src in the zebrafish epithelia. Cells with hyperactive Src may be eliminated through cellular senescence because these signalling pathways are also involved in senescence induction[60]. In contrast, zebrafish

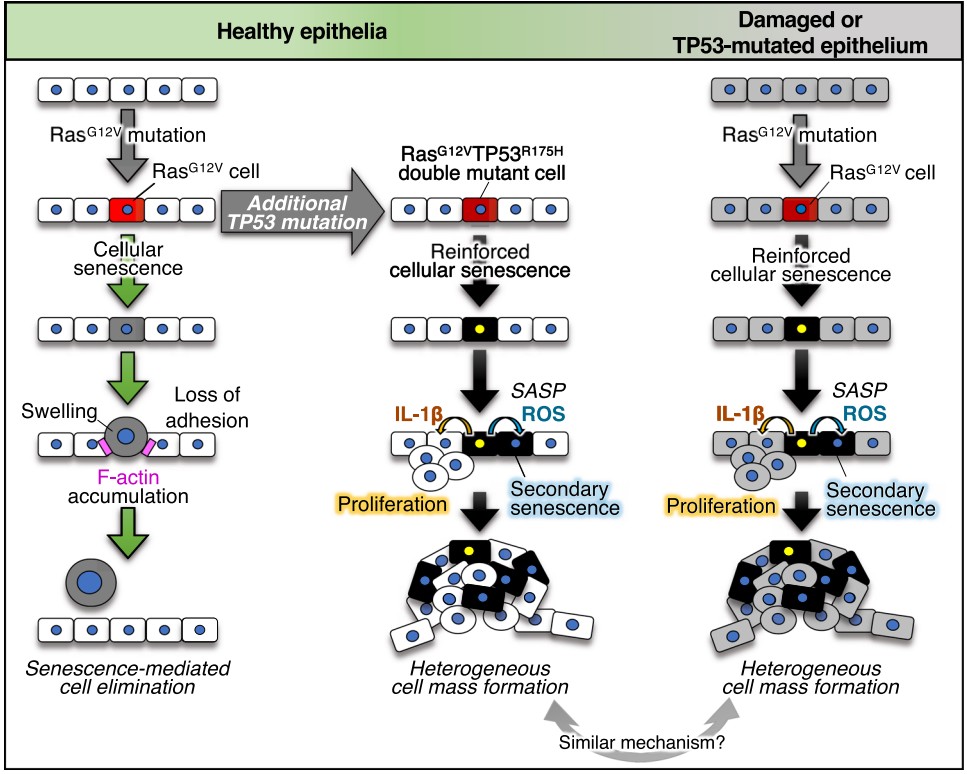

**Fig. 7 Schematic illustration of the roles and regulation of cellular senescence during primary tumorigenesis.** A newly emerged oncogenic cell becomes senescent and is eliminated from the epithelia. Additional TP53 mutation and tissue damage switches oncogene-induced senescence from suppressor to driver during primary tumorigenesis.

epithelia could not eliminate oncogenic cells with TP53 hot-spot mutations (TP53$^{R175H}$, TP53$^{R248W}$ and TP53$^{R273H}$), high Myc levels, and abnormally high Wnt/β-catenin activity[25], suggesting that this elimination system functions against a specific type of oncogenic activation.

Our study also showed that a newly generated Ras$^{G12V}$ cell stimulates senescence neighbouring normal cells, indicating that communication with surrounding cells is involved in Ras$^{G12V}$ cell elimination. Notably, mammalian epithelia can also eliminate sporadically emerging oncogenic cells by communicating with neighbouring normal cells; in Madin-Darby canine kidney cell culture, mosaically introduced oncogenic cells with abnormal Ras and Src activities were apically extruded through communication with neighbouring normal cells[61,62]. This neighbouring cell-mediated oncogenic cell elimination system is known as cell competition or epithelial defence against cancer (EDAC)[63]. A similar phenomenon occurs in cultured cells in the mouse small intestine, pancreas, and lung[64,65]. The mechanisms of EDAC are relatively well-studied. For example, recent studies showed that neighbouring cell-mediated upregulation of pyruvate dehydrogenase kinase in Ras$^{G12V}$ cells and the calcium wave, which is propagated from Ras$^{G12V}$ cells to surrounding cells via the mechanosensitive calcium channel and IP3 receptor, promote Ras$^{G12V}$ cell elimination[26,64]. Interestingly, we found that treatment with 2-aminoethoxydiphenylborane (inhibitor of mechanosensitive calcium channel and IP3 receptor) or dichloroacetate (specific inhibitor of pyruvate dehydrogenase kinase family) significantly suppressed cellular senescence in Ras$^{G12V}$ cells and their elimination (Supplementary Fig. 13a, b), suggesting that these EDAC mediators function upstream of Ras$^{G12V}$ cell senescence and that senescence-driven Ras$^{G12V}$ cell elimination in zebrafish is mediated by mechanisms similar to that of EDAC in mammals. In addition, our findings suggest that cellular

senescence-mediated elimination functions in mammalian epithelial tissues.

We identified membrane E-cadherin localization as a key factor in mutant cell elimination. Mosaically introduced Ras$^{G12V}$ cells blocked membrane localization of E-cadherin in a cell-nonautonomous manner, which drives apical extrusion of Ras$^{G12V}$ cells from the zebrafish larval skin. Additional TP53 mutation recovers membrane E-cadherin and allows retention of mutant cells. However, the detailed mechanisms of how Ras$^{G12V}$ and TP53 mutations control E-cadherin localization were unclear. Notably, Ras$^{G12V}$ cells induced F-actin accumulation in its surrounding cells, whereas Ras$^{G12V}$-TP53$^{R175H}$ double-mutant cells did not, indicating that the levels of F-actin accumulation in neighbouring cells are inversely correlated with E-cadherin levels in mutant cells. One possibility is that F-actin accumulation in neighbours is involved in regulating E-cadherin in the mutant cells. It will be interesting to study the neighbouring cell's machinery controlling E-cadherin in mutant cell and its elimination.

We also showed that inside the growing cell mass, some double-mutant cells and neighbouring cells had reduced E-cadherin levels, whereas the remaining cells did not. It remains unclear how this E-cadherin heterogeneity is induced. Previous reports showed that a SASP factor, including ROS, can disrupt the membrane-localization of cadherin[66]. ROS, which are continuously produced inside the cell mass, may be involved in reducing E-cadherin levels in this context.

Addition of a TP53 gain-of-function mutation prevented Ras$^{G12V}$ cell elimination by restoring intercellular adhesion with neighbours and blocking F-actin accumulation in neighbours to reinforce Ras$^{G12V}$ cell senescence and SASP factor production, resulting in tumour-like cell mass formation in zebrafish epithelia. Ras and TP53 mutations are common genetic mutations in

various cancers, and human pancreatic ductal adenocarcinoma and colorectal cancer cells acquire both mutations during tumour progression[67,68]. Interestingly, recent studies revealed that human TP53 hot-spot point mutations, including TP53[R175H], TP53[R248W], and TP53[R273H], are gain-of-function[69] and enhance oncogenic potential[70,71]. In fact, mice with both Ras and TP53 gain-of-function mutations efficiently form more malignant tumours compared to in mice with either Ras or TP53 mutation[3,7,72,73]. Notably, a previous study reported that an additional TP53[R175H] mutation prevents the disappearance of senescent cells with oncogenic Ras mutations and stimulates tumorigenesis in the mouse pancreas[74]. Although this previous study did not examine the mechanisms of senescent cell disappearance and involvement of SASP in tumorigenesis, the mechanisms may be similar to those we observed in the zebrafish model and in mammals.

We observed a new mode of Ras-TP53-mediated primary tumorigenesis. TP53 mutation stimulates senescent Ras[G12V] cells not only to survive but also to induce proliferation and secondary senescence in the surrounding area. In mouse liver and human fibroblast cultures, senesced oncogenic cells also induced secondary senescence in neighbouring cells[75]. In addition, in human patient specimens, signs of senescence, including oxidative damage accumulation and TP53 upregulation, were detected in normal tissues adjacent to the tumours[76–81], suggesting that secondary senescence occurs in human tumorigenesis. However, the role of secondary senescence in tumorigenesis has not been examined. Here, we demonstrated that secondary senescence in the neighbouring cells of oncogenic cells is required for tumour initiation in a zebrafish model. Thus, our work provides insight into the interactions between oncogenic cells and their surrounding cells during the early stages of cancer development.

Human tumours generally exhibit heterogeneity and include a diverse collection of cells harbouring distinct molecular signatures[82]. Although cancer heterogeneity is the largest barrier to cancer treatment, mainly because of its resistance to therapy[83], the mechanisms that generate heterogeneity in cancer have not been completely explained. The currently acceptable view is that differentiation and/or additional mutations in cancer cells generate tumour heterogeneity[82]. In our zebrafish model, mutant cells converted neighbouring cells to abnormal proliferative or senescent cells through the secretion of SASP factors, thereby forming a heterogeneous cell mass. Based on these findings, SASP-mediated propagation of senescence and abnormal proliferation in the neighbourhood may be a new model of heterogeneous tumour generation. Thus, by using zebrafish imaging analyses, we revealed potential mechanisms of primary tumorigenesis.

## Methods

**Zebrafish maintenance.** Zebrafish wild-type strain (AB), Tg(krt4p:gal4; UAS:EGFP)[27], Tg(krt4p:gal4; UAS:mKO2), Tg(UAS:GAP43mKO2-T2A-H-Ras[G12V]), Tg(UAS:GAP43EGFP-T2A-TP53[R175H]), Tg(cdkn2a/b-hs:Achilles), Tg(UAS:zE-cadherin-GFP), and Tg(il1b:EGFP)[84] were raised and maintained under standard conditions. All experimental animal care was performed in accordance with institutional and national guidelines and regulations. The study protocol was approved by the Institutional Animal Care and Use Committee of the respective universities (Gunma University Permit# 17-051; Osaka University, RIMD Permit# R02-04). One-cell stage embryos were used for cell injection to generate transgenic fish lines or mosaic larvae, with the latter processed for up to 2 dpf. To avoid excess tumour formation, zebrafish larvae were sacrificed within 24 h after a day after the initial cell mass formation, in our study.

**Plasmids.** To prepare UAS promoter-driven plasmids, the UAS promoter was sub-cloned into the pTol2 vector (a gift from Dr. K. Kawakami). Subsequently, membrane-tagged (GAP43-fused) GFP (or GAP43mKO2, mCherry) and T2A were sub-cloned into pTol2-UAS promoter plasmids. These plasmids expressed GAP43GFP (or GAP43mKO2, mCherry) alone in response to Gal4 activation. To generate plasmids expressing such fluorescent proteins with oncogenic proteins,

PCR-amplified cDNAs encoding oncogenic proteins were sub-cloned into the downstream site of T2A of pTol2-UAS-GAP43GFP (or GAP43mKO2, mCherry)-T2A plasmids. Oncogenic protein cDNA were obtained as follows: an oncogenic constitutively active mutant of human H-Ras (Ras[G12V]), in which Gly12 was substituted to Val, was a gift from Dr. A. Yoshimura; Rous sarcoma virus src (v-Src), which is a constitutively active Src kinase, was a gift from Dr. Y. Fujita[62]; human c-Myc is a gift from Dr. W. El-Deiry (Addgene #16011)[85]; zebrafish il1b is a gift from Dr. A. Kawakami[84]; human TP53[WT] and TP53[R175H] was a gift from Dr. B. Vogelstein (Addgene #16543, #16544)[86]; TP53[R248W] and TP53[R273H] and TP53[DD] (TP53 p53 aa 1–11, 305–393)[43] were generated by PCR; transactivation deficient mutants of TP53[R175H] (TP53[QSQS]), in which Leu22, Trp23, Trp53, and Phe54 were substituted to Gln, Ser, Gln, and Ser, respectively[44,45], were established in the lab. TP53 luciferase reporter with p53 binding sites, PG13-luc, was a gift from Dr. B. Vogelstein (Addgene # 16442)[87]. Zebrafish E-cadherin-GFP fusion gene was a gift from Dr. E. Raz[88]. IL-1β5′ untranslated regions were annealed by DNA oligos and cDNA for GFP were PCR-amplified, and these two DNA fragments were cloned into the multi-cloning site of the pCS2p+vector by In-Fusion® HD Cloning Kit (Takara, Kusatsu, Japan). cDNA for α-Bungarotoxin (a gift from Dr. S. Megason, Addgene #69542)[89] was also PCR-amplified and cloned into the multi-cloning site of the pCS2p+vector. Probes for in situ hybridisation were prepared from templates encoding GFP[90], mKO2, zebrafish-codon-optimized Achilles, lcp1[91], npsn[92], il1b, il11b, il6, cxcl8a, and cdkn2a/b and specific primers are described in Supplementary Table 1.

**Generation of transgenic zebrafish.** To generate Tg(UAS:G43mKO2-T2A-H-Ras[G12V]), Tg(UAS:G43EGFP-T2A-TP53[R175H]), or Tg(UAS:zE-cadherin-GFP), each plasmid DNA along with Tol2 transposase mRNA were co-injected into one-cell stage wild-type zebrafish (AB) embryos.

**Mosaic introduction of oncogenic cells.** UAS promoter-driven plasmids (plasmid doses were as follows: fluorescent protein alone; 7.5–15 pg, mosaic Ras[G12V]: 12.5–25 pg, TP53[WT]: 15 pg, TP53[R175H]: 35–50 pg (except for Fig. 3e where 15 pg was used), TP53[R248W]: 35–50 pg, TP53[R273H]: 35–50 pg, TP53[R175H-QSQS]: 35–50 pg, TP53[DD]: 35–50 pg, zebrafish Il1b: 20 pg) were injected into one-cell-stage embryos and maintained at 25.5–28.5 °C until 26–28 h post-fertilisation (hpf) (1 dpf). To immobilise zebrafish larvae, we co-injected α-Bungarotoxin mRNA (10 pg)[89].

**mRNA and antisense oligo MO microinjection.** Capped mRNA was synthesised using the SP6 mMessage mMachine kit (Ambion, Austin, TX, USA) and purified using Micro Bio-Spin columns (Bio-Rad, Hercules, CA, USA). We injected synthesised mRNA at the one-cell stage of zebrafish embryos. To perform knockdown experiments in zebrafish embryos, antisense oligo MOs (Gene Tools, Philomath, OR, USA) were injected into one-cell and/or two-cell stage embryos. MOs against IL-1β (il1b) (5 ng), pu.1 (spi1b) (5 ng)[33], tp53 (2.5–5 ng)[93], and cdkn2a/b (2.5 ng)[94] were used. MO sequences were shown in Supplementary Table 1.

**Chemical treatment.** To evaluate the effects of ROS on cell mass formation, zebrafish larvae were treated with 150 μM N-acetyl-L-cysteine NAC (Sigma-Aldrich, St. Louis, MO, USA) at 18 hpf (approximately 18 somite) for 8 h at 25.5–26.5 °C. For ROS detection (described below), 150 μM NAC solution was added at 10 hpf (approximately bud stage) overnight at 25.5–26.5 °C.

Doxorubicin hydrochloride (FUJIFILM Wako, Osaka, Japan; 040-21521) was dissolved in phosphate-buffered saline (PBS) at 5 mM and stored at -30 °C. Doxorubicin (50 μM) was added at 13 hpf (approximately 10 somite stage) for 10 h at 25.5–26.5 °C. Hydrogen peroxide (300 μM) was added at 13 hpf (approximately 10 somite stage) for 10 h at 25.5–26.5 °C.

2-Aminoethoxydiphenylborane (Sigma-Aldrich; D9754) was dissolved in dimethyl sulphoxide (DMSO) at 150 mM and stored at −30 °C. 2-Aminoethoxydiphenylborane (6.25 μM) was added at 18 hpf (approximately 18 somite) (Supplementary Fig. 13a) or at 24 hpf (approximately Prim-6 stage) (Supplementary Fig. 13b) at 28.5 °C. Dichloroacetate (Selleck Chemicals, Houston, TX; S8615) was dissolved in sterilized distilled water at 150 mM and stored at -30 °C. Dichloroacetate (30 mM) was added at 18 hpf (approximately 18 somite) (Supplementary Fig. 13a) or at 24 hpf (approximately Prim-6 stage) (Supplementary Fig. 13b) at 28.5 °C.

**Detection of ROS.** To detect ROS production, zebrafish larvae were incubated with 2.5 μM CellROX DeepRed (Thermo Fisher Scientific, Waltham, MA, USA) in egg water for 2 h, or with 5 μM CellROX Green (Thermo Fisher Scientific) in egg water for 5 h at 25.5-26.5 °C. After loading, embryos were washed three times in egg water and confocal images were obtained.

**Detection of SA-β-gal-positive senescent cells using SPiDER-βGal.** SA-β-gal using SPiDER-βGal was performed according to the manufacturer's instructions. In detail, larvae were fixed with 4% paraformaldehyde in PBS (4% PFA) for 1 h at RT. After washing three times with PBS, to detect senescent cells, larvae were incubated with SPiDER-βGal (DOJINDO, Kumamoto, Japan, SG03), which enables specific detection of SA-β-gal activity-positive cells in living tissue at single-

cell resolution[95], in McIlvaine buffer (pH 6.0, dilution 1/1000) for 30 min at 37 °C. After loading, larvae were washed three times in 0.1% Tween 20 in PBS, and then immunostaining was performed.

**Detection of proliferating cells using EdU.** To detect proliferating cells, zebrafish larvae were incubated with 500 μM EdU/10% DMSO in Egg water for 1 h, and larvae were washed three times in egg water. Larvae were fixed with 4% PFA in PBS for 1 h at RT. The larvae were rinsed in double-distilled water, placed in acetone at 20 °C for 7 min, rinsed in double-distilled water, and permeabilized for 1 h in PBS/1%DMSO/1%Triton. For EdU detection, larvae were processed according to the Click-iT EdU Cell Proliferation Assay Kit (Invitrogen, Carlsbad, CA, USA; Alexa Fluor 488: C10637 and Alexa Fluor 647: C10640). After loading, larvae were washed three times in 0.1% Tween 20 in PBS, and then immunostaining was performed.

**Whole-mount immunostaining.** At 26–32 hpf, Tg(krt4p:gal4) larvae with mosaically introduced fluorescent cells were manually dechorionated and then fixed with 4% PFA overnight at 4 °C. As for H3K9me3, antigen retrieval was performed[96]. The larvae were washed four times with 0.5% Triton X-100 (PBST) and incubated with blocking reagent (10% foetal bovine serum, 4% Block Ace (Megmilk Snow Brand, Tokyo, Japan), and 1% DMSO in 0.1% PBST) for 1 h (γH2AX-stained larvae were incubated with blocking reagent without Block Ace). The larvae were incubated with primary antibodies overnight at 4 °C, washed, and incubated with AlexaFluor-conjugated secondary antibodies and with Hoechst33342 (Invitrogen) overnight at 4 °C. Immunostained larvae were visualised with a M205FA fluorescent stereomicroscope (Leica), LSM700, or FV1000 (Olympus, Tokyo, Japan) confocal laser-scanning microscope. Images were prepared and analysed using ImageJ software (NIH, Bethesda, MD, USA). A list of primary and secondary antibodies is shown in Supplementary Table 1.

**Whole-mount in situ hybridisation.** Whole-mount in situ hybridisation was performed according to a standard protocol. Fluorescence in situ hybridisation was performed according to a previously described protocol[97]. Digoxigenin- or FITC-labelled RNA antisense probes were prepared from plasmids containing *GFP*, *Achilles*, *mKO2*, *lcp1*, *npsn*, *il1b*, *il6*, *il11b*, *cxcl8a*, and *cdkn2a/b*. Images were taken using a M205A stereomicroscope (Leica) and FV3000 confocal laser scanning microscope.

**Haematoxylin and eosin staining.** The fixed larvae were embedded in paraffin and serially sectioned. Sections of 4-μm thickness were stained with haematoxylin and eosin according to a standard procedure.

**Estimation of cytoplasm and nuclear size.** The cytoplasm/nuclear areas in each cell were measured using ImageJ software. To estimate the size of the cytoplasm/nuclear area, the sum of the measured areas was multiplied by the thickness between the slices.

**Quantification of membrane zebrafish E-cadherin-GFP levels.** In Supplementary Fig 5a, b, UAS:zE-Cadherin-GFP plasmid (10 pg) were co-injected with UAS:mCherry-T2A-H-Ras[G12V] into one-cell-stage Tg embryos (krt4p: gal4). In Supplementary Fig 5c, d, UAS:mCherry-T2A-H-Ras[G12V] were injected into one-cell-stage Tg embryos (krt4p: gal4; UAS:zE-cadherin-GFP). The fluorescence intensity of E-cadherin-GFP in the membrane and cytosolic areas in each region was measured after subtracting background signals using ImageJ software (NIH, Bethesda, MD, USA).

**Detection of F-actin.** To detect F-actin, 1 dpf zebrafish larvae were incubated with Alexa Fluor488-conjugated phalloidin (Invitrogen, Carlsbad, CA, USA; A12379, 1:50 dilution) overnight at 4 °C.

**Quantitative PCR (qPCR).** UAS promoter-driven plasmids (Ras expression plasmid doses were as follows: 25 pg in Supplementary Fig. 3c, e, 15 pg in Supplementary Fig. 7d, 12.5 pg in Supplementary Fig. 12 f) were injected into one-cell-stage embryos. Total RNA from 25 larvae was purified using TRIzol reagents (Invitrogen), and cDNAs were synthesised using ReverTra Ace qPCR RT Master Mix (Toyobo, Osaka, Japan). qPCR was performed on an Mx3000P QPCR system (Agilent Technologies, Santa Clara, CA, USA) with THUNDERBIRD SYBR qPCR Mix (Toyobo); the specific primers are described in Supplementary Table 1; *actb1* was used as a loading control. qPCR cycling conditions were as follows: 95 °C for 1 min, [95 °C for 10 s, 60 °C for 30 s] (45 cycles), followed by dissociation curve analysis. The primer list is shown in Supplementary Table 1.

**Generation of Tg(cdkn2a/b-hs:Achilles) zebrafish.** To generate a donor plasmid containing a heat-shock promoter and cDNA encoding zebafish-codon-optimized and nuclear-localization-signal-tagged Achilles (Tbait-hs-zAchilles-NLS), zAchilles-NLS cDNA was synthesised (FASMAC, Atsugi, Japan) and replaced with GFP sequence in Tbait-hs-GFP plasmid (a gift from Dr. S. Higashijima)[98]. Tbait-hs-zAchilles-NLS was co-injected with a single-guide RNA (sgRNA) targeted for digestion of *cdkn2a/b*, an sgRNA targeted for donor plasmid digestion, and Cas9 mRNA into one-cell stage wild-type embryos. A transgenic fish was outcrossed with a wild-type fish to produce the founder line. sgRNA sequences are described in Supplementary Table 1.

**Luciferase assays.** HEK293 cells were grown in Dulbecco's modified Eagle's medium (DMEM) supplemented with 10% fetal bovine serum and a 100 U/ml penicillin-streptomycin mixed solution. Cells were seeded into 35 mm-diameter plates and transfected with the expression plasmids and TP53 reporter plasmids using polyethylenimine MW 25000 (Polysciences, Warrington, PA, USA). After 24 h, firefly and *Renilla* luciferase activities were determined with the Promega Dual Luciferase Assay System (Promega, Madison, WI, USA). The pRL-EF vector, which expresses *Renilla* luciferase under the control of the EF-1α promoter, was used to normalize the transfection efficiency of the luciferase reporters. The values shown are the averages of experiments repeated three times, with each transfection performed in duplicate for each experiment.

**Time-lapse imaging.** For time-lapse live imaging, Tg(krt4p:gal4) larvae with mosaically introduced Ras[G12V] were manually dechorionated using forceps and then mounted in 2% methylcellulose onto glass-bottom dishes (Matsunami Glass, Osaka, Japan). Images were collected using an MZ16FA stereomicroscope (Leica, Wetzlar, Germany) equipped with Objective plan-apo 1.0x (10447157) at 28 °C for approximately 2 h, 30 min, or 6 h (Supplementary Movie 1, 2, or 3, respectively). The time interval was 5 min.

For time-lapse confocal live imaging, larvae were manually dechorionated by forceps and mounted in 1% low melting agarose with egg water onto glass-bottom dishes. Live imaging was performed using an FV3000 confocal laser scanning microscope (Olympus, Tokyo, Japan) equipped with a 20 × 1.0 NA objective. Two laser lines, 488 and 561 nm were used. The recording interval was 20 min. At each time point, 40–50 confocal slices through the z-axis were acquired. Collected image data were processed using ImageJ.

**Statistics and Reproducibility.** Differences between groups were examined using a two-tailed unpaired Student's *t*-test, one-way analysis of variance, and Kruskal-Wallis test in Excel (Microsoft, Redmond, WA, USA), R (The R Project for Statistical Computing), or Prism8 (GraphPad, Inc., San Diego, CA, USA). *p* values < 0.05 were considered to indicate significant results. Figures of representative images or plots were reproduced in at least two (Figs. 2a–e, 4a, 5c–i, 5k, 6a–i, Supplementary Fig. 1e, 3a–e, 4b, 5b–d, 6b–c, 7c–d, 8a, 9a–b, 10b, 11a–c, 12a–d, 12f, 13a–b), or three or more (Figs. 1b, c, 2f, 3a–d, 4b, 5a–b, 5j Supplementary Fig. 1a–d, 2a–e, 4a, 5a, 6a, 7b, 8b, 10c–f, 11d, 12e) independent experiments.

**Reporting summary.** Further information on research design is available in the Nature Research Reporting Summary linked to this article.

## Data availability
Source data are provided with this paper. All the other data are available within the article and its Supplementary Information. Source data are provided with this paper.

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

## Acknowledgements

We thank E. Hara for his helpful discussion, H. Wada, K. Kawakami, A. Yoshimura, B. Vogelstein, W. El-Deiry, Y. Fujita, E. Raz, S. Higashijima, and A. Miyawaki for providing the plasmids, H. Wada and NBRP for providing transgenic zebrafish, and Ishitani lab members for their helpful discussion, technical support, and fish maintenance. This research was supported by the Takeda Foundation (T.I.), Mitsubishi Foundation (T.I.), Kao Foundation (Y.A.), Nagase Science and Technology Foundation (T.I.), and Kawano Foundation (T.I.), Daiichi Sankyo Foundation (T.I.), Uehara Memorial Foundation (T.I.), Mochida Memorial Foundation (T.I.), AMED (21gm5010001h0005) (T.I.), and a Grant-in-Aid for Scientific Research on Innovative Areas (25117720, 20H05365) (T.I.) (20H05534) (Y.A.), Young Scientists (20770514) (Y.A.), Exploratory Research (20K21502) (T.I.), Transformative Research Areas(A) (21H05287) (T.I.), JST SPRING (Y.H.), and Scientific Research (B) (19H03412) (T.I.).

## Author contributions

Conception and design: Y.H., Y.A., and T.I.; Investigation: Y.H., Y.A., Y.N., C.M., and T.I.; Writing and review: Y.H. and T.I.; Writing contribution and review: Y.H., Y.A., and T.I.

## Competing interests

The authors declare no competing interests.
