## [Peer Review File · Nature Communications]

Zebrafish imaging reveals TP53 mutation switching oncogene-induced senescence from suppressor to driver in primary tumorigenesisReviewers' comments:

Reviewer #1 (Remarks to the Author):

The manuscript entitled "Zebrafish imaging reveals additional TP53 mutation that switches oncogene-induced senescence from suppressor to driver in primary tumorigenesis" by Haraoka and collaborators describes how a gain-of-function mutation in p53 induces a switch in oncogenically-induced senescent cells from tumor suppressive to tumor promoting. The authors report that mosaic activation of oncogenic Ras (RasG12V) in the skin of zebrafish larvae leads to induction of senescence and elimination of these damaged cells, while widespread activation of RasG12V is inconsequential. In contrast, the combination of RasG12V activation and a gain-of-function p53 (R175H) leads to the induction of a "stronger" senescence response that affects the surrounding cells through secreted factors, and this leads to tumour-like formation. These findings are, in the end, substantiated by the observation of the protumorigenic contribution of a senescent ZF when RasG12V expressing cells are introduced in previously damaged larvae in the absence of p53 mutation.

The manuscript raises a number of intriguing questions and presents thought provoking results. The implications of the findings are substantial but require further detail interrogation.

A general concern is related with the way in which senescence is identified. Why the authors did not attempt SA-beta-Galactosidase staining which, despite many objections, is the most widely accepted marker of senescence? Similarly, proliferation would also be very helpful to identify senescence growth arrest.

Introduction of RasG12V cells in normal skin results in senescence of the RasG12V expressing cells, but not of the neighbouring cells? What are the surrounding cells providing the RasG12V cells to allow senescence induction?

Ras-expressing cells show decreased membrane E-cadherin and this is suggested to be behind the elimination of these cells. This does not occur when p53 or cdkn2a/b are suppressed with MO. How about ubiquitous expression of RasG12V, do these cells show membrane E-cadherin? What about the double RasG12Vp53R175H?

What happens with ubiquitous expression of RasG12Vp53R175H?

Other oncogenes, such as src, show a similar effect, but not others, such as Myc. What about other p53 gain-of-function mutants like R273H and R248W?

The authors claim that the introduction of a p53R175H mutant in RasG12V expressing cells leads to a "stronger" senescence induction, while complete abrogation of p53 does not. They measure senescence by gammaH2AX foci formation and cdkn2a/b expression. They also report that these cells produce "SASP" and they claim that this is represented by il1b, il11b and ROS. To my knowledge, ROS is not universally considered a SASP factor. All these factors are also produced by RasG12V expression alone, is the different effect just caused by a higher level of induction of these factors? What happens to cdkn2a/b after complete abrogation of p53 in RasG12V expressing cells? Il1b and ROS are pointed out as responsible for tumour formation since MO against il1b and ROS scavenger NAC limit tumour mass formation. Could the authors recapitulate tumour mass production by RasG12V expression combined with il1b or increased ROS?

One possible alternative explanation for the observations is that RasG12V-expressing cells are unstable and are eliminated, while p53 mutant provides a survival activity to RasG12V cells, allowing them to produce and secrete these factors for a longer period of time, leading to tumour formation. Have the authors checked apoptosis in their systems?

Then the authors start a series of experiments and claims that to this reviewer are confusing. On the one hand, they claim that RasG12V-p53R175H cells induce features of senescence in neighbouring cells, such as ROS production, but at the same time they claim they induce proliferation, as measured by phosphor-histone H3. Could the authors further clarify this issue? Similarly, they claim that the tumour masses contain cells that show markers of what they consider senescence (although this claim is not supported by proliferation markers, for example) and at the same proliferate to form the tumour. And they might carry RasG12V-p53R175H or not. Double mutant RasG12V-p53R175H do not produce tumour masses in p53KD ZF skin and they claim that this means that induction of paracrine senescence is required for tumour formation. Have the authors checked what happens when they introduce RasG12V alone in p53KD skin? Authors also show that ROS and il1b are uniformly expressed throughout the tumour masses and claim that this means that senescence is propagated by senescent neighbouring cells, which is a

speculative claim difficult to reconcile with the proliferation required to form a tumour. Introduction of RasG12V expressing cells in mutant p53R175H skin leads to the formation of tumour masses. This would imply that now RasG12V expression alone does not lead to elimination of these cells and results in “strong” senescence. Since mutant p53 is not present in the same cell than mutant Ras, this implies that expression of some factor/s is affecting Ras expressing cells. Have the authors identified this factor/s? Have they tried to decrease il1b in p53R175H mutant cells or block ROS production?

Finally, the authors use damaged epithelia (doxorubicin treated ZF) to claim that a senescent tissue does not retain the ability to eliminate RasG12V expressing cells and this leads to tumour formation. Do RasG12V expressing cells show increased cdkn2a/b, il1b, il11b or ROS? What is the effect of eliminating p53 in RasG12V expressing cells in the context of doxo-induced senescent skin? Would doxo treatment of p53null skin promote tumour formation by RasG12V expressing cells?

Reviewer #2 (Remarks to the Author):

Haraoka and colleagues used the zebrafish larval skin model to study the response of epithelial cells to oncogene expression. In their work, they make two major claims: (i) oncogene-induced senescence leads to clearance from the epithelium independently of the immune system; (ii) concomitant expression of oncogenic Ras and mutant p53 lead to the paradoxical potentiation of senescence, leading to increased proliferation and senescence in adjacent normal tissue and the ultimate formation of a tumor-like mass.

The authors present an interesting account of the response of normal cells to oncogenes using an elegant experimental system. And while their account adds to a growing line of research that highlights intrinsic mechanisms of tumor suppression and potentiation embedded in tissue architecture and cell-cell interactions (see for example Brown S et al. 2017), there are several aspects of the study that are not convincing or overinterpreted. A substantial effort would be required to provide data that actually support the authors' claims.

Major points:

The authors claim that mutant p53 increases the frequency of senescent cells, leading to proliferation of neighboring cells due to SASP. However, not every Ras/p53 mutant cell has senescent markers. Therefore, one interpretation is that mutant p53 is diversifying the population in two ways: (i) increasing proliferation in a non-senescent subpopulation and (ii) increasing hallmarks of senescence in another one. This idea is conceptually reminiscent of a model advanced by Sean Morrison's group (<https://www.nature.com/articles/nature12830>) where they claimed that oncogenic Nras can increase proliferation of a subpopulation of HSPCs, and increase self-renewal of another subpopulation. Other studies have shown that the induction of senescence in one cell type can induce the reprogramming of an adjacent cell (<https://science.sciencemag.org/content/354/6315/aaf4445.long>).

The authors co-introduce the TP53R175H mutation with Ras and suggest this reinforces senescence based on the induction of some damage associated markers, cdkn2a/b, and the expression of inflammatory factors also linked to the SASP. There are a number of problems with this interpretation. For example, Il1b could be induced in any inflammatory setting, and ROS is by no means senescence specific. Certainly while cdkn2a is induced during senescence in mammalian systems, inactivation of p53 leads to aberrant activation of the locus (so here high cdkn2a levels are actually indicative of failure of the program). While having good markers of the senescence program have limited the field, there is so little characterization of the cells in the zebrafish model that the authors' bold claims relating to the biology are preliminary.

Related to the above point, there is a paucity in data regarding cell proliferation in oncogene expressing cells, despite the fact that this is one of the classical phenotypic outcomes attributed to both oncogenes and senescence. Are oncogene-expressing cells dividing or not? Does cdkn2a/b activation correlate with decreased division (in the background of oncogenic Ras, or Ras+TP53)? Are dividing or arrested cells the ones that are producing SASP? The answers to these questions are important to understand which cells are contributing directly or indirectly to the growth of a tumor-like mass.

Figures 3D and 5F present very similar experiments, but one is used to imply that p53 knockdown in oncogene expressing cells is insufficient to generate a tumor-like mass, whereas the other one is used to conclude that p53-dependent secondary senescence is important for tumor growth. Perturbation of p53 specifically in oncogene-expressing cells would be necessary to disentangle these two experiments and conclusions.

The mechanism of oncogene expressing cell elimination is not clear though as the authors suggest it does not appear to be immune mediated. The authors further state that their data "indicates that cellular senescence can suppress tumorigenesis through immune cell-independent oncogenic cell elimination". All of the data here are correlative and there is no direct evidence for this statement.

The data interpretation of the results that p53R175H acts through a gain of function are not decisive and difficult to interpret. As I understand the experiment the results in otherwise p53 wt background, so p53 could simply be activating as a dominant negative. While apparent knockdown of wt p53 in that system does not enhance tumor growth, but we don't know whether there could be simply quantified differences on p53 output. More decisive comparisons (and evaluation of output) would be needed to relate this to p53 'gain of function'. Of course then one would wonder how that actually worked.

Reviewer #3 (Remarks to the Author):

While tumors are thought to arise from oncogenic cells that acquire additional mutations, the mechanisms by which these additional mutations promote tumorigenesis is not well understood. The manuscript by Haraoka et al explores how additional mutations in oncogenic cells may promote oncogene-induced senescence and become a driver of tumorigenesis. The authors utilize the larval zebrafish to tackle this question, a system that facilitates rapid in vivo imaging based analysis after genetic manipulation. The authors find that the generation of oncogenic KRAS+ cells in otherwise healthy epithelia stimulates their elimination from the tissue. While previous work has focused on cellular senescence as a tumor suppressive mechanism, the authors find that the acquisition of TP53 gain of function mutations in combination with oncogenic KRAS suppresses elimination from the tissue and promotes a senescence-associated secretory phenotype (SASP). The production of SASP factors in the KRAS/TP53 R175H oncogenic cells converts neighboring healthy cells into senescent cells, induces the production of SASP factors and promotes the generation of a cell mass that resembles a human tumor. Together, the authors conclude that oncogene-induced senescence may occur at early stages of human tumorigenesis. This is an interesting paper, and the findings further our understanding of the initial events occurring in and around oncogenic cells to promote tumorigenesis. Thus, these potentially important findings would be of broad interest to the scientific community. However, several key aspects of the story are not fully supported by the data and need to be strengthened to be convincing.

Major Points

1) It seems that all of the oncogenic cells are present in the head region of the zebrafish larvae. Is this the case? It would be helpful to clarify if there is something specific about this area, or did the authors just choose to focus on this region for analysis. Likewise, it is not clear where on the animal that the cell masses formed. Presumably, it is in the same area? This should be mentioned in the text, and potentially illustrated with a schematic.

2) For several quantifications, the legend states that each dot represents an individual embryo. Yet, the number of embryos analyzed seems low given the high-throughput nature of the system (Fig 1 B, 2D, 3A). Likewise, it is not clear if these embryos came from three independent experiments. Additional analysis should be performed to bolster these claims and information should be added to the figure legends to clarify if embryos from independent experiments were analyzed.

3) The authors state that "The oncogenic cells appear to occur in an immune cell-independent

manner". While the authors use *spi1b* to suppress myeloid lineage formation, no controls are provided to show this perturbation was effective. It would be important to show macrophages and/or neutrophils are normally present at this time in development and are significantly depleted by the *spi1b* MO to support their claims.

4) This also brings up the interesting question of how the oncogenic cells are eliminated. The authors cite a recent paper from Takeuchi et al that provides mechanistic insight into how epithelial cell extrusion is regulated to clear unfit cells. Can the authors comment if that is the case here? Along these lines, Figure 1c nicely shows rounded KRAS⁺ cells, but it would be useful to show neighboring healthy GFP⁺ cells as well to fully demonstrate that the KRAS cell is out of the plane of tissue, and if it is actively being eliminated by the epithelial neighbors.

5) Are the KRAS⁺ cells undergoing apoptosis (i.e., positive for cleaved caspase 3 or TUNEL) during elimination from the tissue? This could provide clues as to the mechanisms used to eliminate these cells from the tissue and may also support the idea that the additional TP53 mutations suppress cell death and drive senescence.

6) The authors frequently refer to KRAS⁺ TP53 R175H double mutant cells. While KRAS is fluorescently tagged, overlap with a fluorescently tagged TP53 R175H is never shown. Given the mosaic nature of injections, expression of these two constructs in the same cell may not always be the case. Alternatively, it could be that activated TP53 in healthy neighbors influences the KRAS cells, a point addressed later in the paper. It would be helpful for the authors to demonstrate an increase in TP53 R175H levels and determine what cell type it is expressed in.

7) On pg 8, the authors state "these results suggest that the addition of the TP53 R175H mutation reinforces RAS G12V induced senescence." It would be useful to show that TP53 loss of function does not reinforce senescence in this context.

8) The authors state that the "effects of the p53 R175H mutation on these (ROS, IL1b and IL11b) levels were minor (Figure 2a, 4a, 4b)". As Figure 2a only shows ROS, to better support this idea, it would be important to show the impact of KRAS alone and TP53 R175H alone on IL1B and IL11B expression specifically within these cells in Figure 4C. Also, is this observed expression increase dependent on ROS levels? Determining if this increase is abolished when suppressing ROS with NAC would help further support this mechanism.

9) For IL1B MO experiments in Figure 5C, an important control would be to show IL1b (but not other SASPs) is no longer induced or that expression is significantly diminished.

10) The authors nicely show the formation of cell masses, but it is not clear whether changes in cell-cell adhesion occur within the cell mass. What is the status of E-cadherin levels in KRAS⁺ TP53 R175H positive cells and their neighbors within the cell mass?

11) The authors argue that the double mutant KRAS p53 R175H mutant cells induce senescence in the neighboring cells, supported by IL1b and ROS being broadly expressed in the cell mass (Supp Fig 3a,c). Based on this result, then one would predict that induced ROS and IL1B expression in the neighboring cells would be suppressed in the TP53 MO skin. Is this indeed the case?

12) The authors use doxorubicin to create senesced epithelia, and while quite intriguing, this new model is not fully characterized making it difficult to interpret the findings. In line with the rest of the manuscript, it would be important to show markers of senescence (i.e., ROS dye and IL1b levels) are increased after Dox exposure. Also, does this treatment induce apoptosis in the epithelial cells with DNA damage, as this could confound the interpretation of the results.

Minor points:

It would be helpful to use the TP53 R175H notation in the figures, as opposed to TP53⁺, which could be interpreted as containing wild type TP53

For consistency and to aid colorblind readers, the red images in Figure 4c should be made magenta

The data in Supp Fig 3 supports the idea that TP53 R175H promotes accumulation of senescent cells to form masses. Given the mechanistic link with ROS and I11b later in the manuscript, including this data in the main figures (especially a-c) would help to focus the message for the reader.

On pg9, the authors state "Surviving double mutant zebrafish skin secreted SASP factors". While the authors nicely show induced expression of SASP factors, the data presented does not demonstrate that these factors are indeed secreted. The authors should change the language to reflect this fact.

Typo, top of pg 10 "arrowhaeds"

Manuscript ID: **NCOMMS-20-35578-T**

Authors: Haraoka et al.

Title: **Zebrafish imaging reveals TP53 mutation switching oncogene-induced senescence from suppressor to driver in primary tumorigenesis.**

The manuscript has been revised in accordance with the comments raised by the four referees. Our responses to their comments are as follows.

Author responses to the comments of Reviewer #1

We thank Reviewer#1 for the careful and constructive review of our paper and for the positive comments on our findings. As indicated in the responses below, we have considered all comments and suggestions in the revised version of the manuscript.

A general concern is related with the way in which senescence is identified. Why the authors did not attempt SA-beta-Galactosidase staining which, despite many objections, is the most widely accepted marker of senescence? Similarly, proliferation would also be very helpful to identify senescence growth arrest.

Response: We appreciate this comment. Based on this suggestion, in the revised manuscript, we have added new data showing SA-beta-galactosidase (SA- β -gal) activity and cell cycle arrest in Ras^{G12V} cells and Ras^{G12V}-TP53^{R175H} double-mutant cells.

To detect SA- β -gal activity, we used SPiDER- β Gal (Dojindo) to specifically detect SA- β -gal activity-positive cells in living tissue at single-cell resolution (Doura et al., *Angew Chem Int Ed Engl.* 2016). SPiDER- β Gal has widely been applied to detect SA- β -gal-positive senescent cells in several studies (Han et al., *Mol Cell* 2018; Barinda et al., *Nat commun* 2020). As shown in revised Fig 2b, SPiDER- β Gal stained Ras^{G12V} cells and the additional TP53^{R175H} mutation enhanced this staining.

To detect cell cycle arrest, we performed EdU staining. As shown in revised Fig 2c, EdU incorporation occurred in control cells and TP53^{R175H} cells, whereas

incorporation was significantly suppressed in Ras^{G12V} cells and Ras^{G12V}-TP53^{R175H} double-mutant cells, indicating that the cell cycle was arrested in almost all of Ras^{G12V} cells and double-mutant cells.

These results support that cellular senescence occurred in Ras^{G12V} cells and that the additional TP53^{R175H} mutation facilitates Ras^{G12V}-induced senescence.

Introduction of RasG12V cells in normal skin results in senescence of the RasG12V expressing cells, but not of the neighbouring cells?

Response: As shown in revised Fig 2a and 2d (corresponding to original Fig 2a and 2b, respectively), the introduction of Ras^{G12V} cells in normal skin induced expression of the senescence markers γ H2AX and *cdkn2a/b* in Ras^{G12V}-expressing cells but not in neighbouring cells. Furthermore, in revised Fig 2b and 5k, we showed that SA- β -gal activity (evaluated by SPiDER- β Gal) was upregulated in Ras^{G12V}-expressing cells but not in neighbouring cells. These new data strongly support our hypothesis.

What are the surrounding cells providing the RasG12V cells to allow senescence induction?

Response: This is an interesting point. We expect that the Ras^{G12V} cell elimination in zebrafish is mediated by mechanisms similar to those in “EDAC”-mediated Ras^{G12V} cell elimination. As mentioned in both the original and revised manuscript, previous studies reported that the mammalian epithelia can also eliminate Ras^{G12V} cells by communicating with surrounding normal cells; this phenomenon is known as EDAC. The mechanisms of EDAC are relatively well-studied. For example, the calcium wave, which is mediated by the mechanosensitive calcium channel and IP3 receptor, is propagated from Ras^{G12V} cells to the surrounding cells and stimulates polarized movement of the surrounding cells toward Ras^{G12V} cells (Takeuchi et al., *Curr Biol* 2020). The surrounding cells also upregulate PDK (pyruvate dehydrogenase kinase) and consequently induce metabolic change in Ras^{G12V} cells (Kon et al., *Nat Cell Biol* 2017). Next, calcium wave-mediated polarized movement of the surrounding cells and PDK-mediated metabolic change promote Ras^{G12V} cell apical extrusion. Inhibiting the calcium wave or PDK by treatment with 2-aminoethoxydiphenylborane (2APB; inhibitor for mechanosensitive calcium

channel and IP3 receptor) or dichloroacetate (DCA; specific inhibitor of the PDK family), respectively, can significantly suppress Ras^{G12V} cell elimination (Takeuchi et al., *Curr Biol* 2020; Kon et al., *Nat Cell Biol* 2017). Therefore, we tested the effects of 2APB and DCA on Ras^{G12V} cell behaviour in zebrafish skin. As shown in revised supplementary Fig 12a and b, treatment with 2APB and DCA significantly suppressed cellular senescence in Ras^{G12V} cells and their elimination, suggesting that calcium wave-mediated cell-cell communication and neighbouring cell-mediated PDK activation are involved in Ras^{G12V} cell senescence. In further studies, we would like to clarify the detailed mechanism by which the surrounding cells stimulate Ras^{G12V} cell senescence.

In contrast to the induction of Ras-induced cellular senescence in cultured cells, which takes more than 6 days (Serrano et al., *Cell* 1997), this cell-nonautonomous cellular senescence in zebrafish skin occurs within one day. This result strongly suggests there is a mechanism that actively promotes cellular senescence in this context. The interaction of Ras^{G12V} cells with neighbouring cells may accelerate Ras-induced senescence. In the future, we would like to clarify how this cell-nonautonomous cellular senescence occurs.

Ras-expressing cells show decreased membrane E-cadherin and this is suggested to be behind the elimination of these cells. This does not occur when p53 or cdkn2a/b are suppressed with MO. How about ubiquitous expression of RasG12V, do these cells show membrane E-cadherin? What about the double RasG12Vp53R175H?

Response: To answer these questions, we analysed the E-cadherin levels in larval skin expressing Ras^{G12V} ubiquitously or Ras^{G12V}-TP53^{R175H} mosaically. First, we found that membrane E-cadherin levels in each cell in larval skin ubiquitously expressing Ras^{G12V} were slightly lower than those in the normal skin but were much higher than those in mosaic Ras^{G12V} cells (revised Supplementary Fig 4b), suggesting that non-cell autonomous mechanisms contribute to downregulating membrane E-cadherin in mosaic Ras^{G12V} cells. We also unexpectedly found that mosaic Ras^{G12V}-TP53^{R175H} cells retained membrane E-cadherin, whereas mosaic Ras^{G12V} cells lost this protein (revised Supplementary Fig 4c). Because cellular senescence mediates membrane E-cadherin reduction in mosaic Ras^{G12V} cells and additional TP53^{R175H} mutation

enhances senescence in Ras^{G12V} cells, we initially expected that membrane E-cadherin levels in mosaic Ras^{G12V}-TP53^{R175H} cells would be low. A cellular senescence-independent mechanism may have recovered the E-cadherin levels in double-mutant cells. For example, the levels of F-actin accumulation in neighbouring cells were inversely correlated with E-cadherin levels in mutant cells (revised Supplementary Fig 1e v.s. Supplementary Fig 4c), suggesting that F-actin accumulation might negatively regulate E-cadherin in the mutant cells. We have briefly discussed this point in the revised manuscript.

Other oncogenes, such as src, show a similar effect, but not others, such as Myc. What about other p53 gain-of-function mutants like R273H and R248W?

Response: We analysed the behaviour of mosaically introduced cells with a TP53 gain-of-function mutation, including TP53^{R175H}, TP53^{R248W}, and TP53^{R273H}. We found that mosaic cells with these mutations were not eliminated (revised Supplementary Fig 1h–j).

The authors claim that the introduction of a p53^{R175H} mutant in Ras^{G12V} expressing cells leads to a “stronger” senescence induction, while complete abrogation of p53 does not. They measure senescence by gammaH2AX foci formation and cdkn2a/b expression. They also report that these cells produce “SASP” and they claim that this is represented by il1b, il11b and ROS. To my knowledge, ROS is not universally considered a SASP factor. All these factors are also produced by Ras^{G12V} expression alone, is the different effect just caused by a higher level of induction of these factors?

Response: Reviewer#1 pointed out that ROS is not universally considered as a SASP factor. However, the review article by Gorgoulis et al. (*Cell* 2019), which presents the consensus from the International Cell Senescence Association (ICSA), described ROS as a SASP factor. Therefore, we considered ROS as a SASP factor.

Reviewer#1 may be concerned that our choice of SASP factors is insufficient. Therefore, we added new data for other typical SASP factors, such as *il6* (IL-6) and *cxcl-8a* (CXCL8/IL-8) (Gorgoulis et al. *Cell* 2019). As shown in revised Supplementary Fig 2b, expression of these SASP factors also synergistically upregulated by Ras^{G12V} and TP53^{R175H} mutations.

Reviewer#1 may also be concerned that the importance of a high level of SASP factor production in double-mutant cells is unclear. As shown in revised Fig 3d, 4a, 5a, 5h, and Supplementary Fig 2c (corresponding to original Fig 3d, 4a, 5c, 5e, and 4b) and revised Fig 4b, Ras^{G12V} and TP53^{R175H} double-mutant cells produced SASP factors, IL-1 β and ROS, at a higher level and induced IL-1 β -dependent proliferation, ROS-dependent senescence in neighbouring cells, and consequent generation of tumour-like cell masses. In contrast, cells with Ras^{G12V} alone produced IL-1 β and ROS at a lower level and did not induce neighbouring cell proliferation and senescence or cell mass formation. Furthermore, in revised Fig 5d, we showed that forced expression of IL-1 β and treatment with ROS (H₂O₂) synergistically induced tumour-like cell mass formation in the larvae mosaically introduced with Ras^{G12V} cells. These results suggest that a high-level production of SASP factors (IL-1 β and ROS) enables neighbouring cell proliferation and senescence and cell mass formation. In addition, we considered that blocking cell elimination by additional TP53 mutation enhanced neighbouring cell proliferation and senescence. Although Ras^{G12V} single-mutant cells express SASP factors, they cannot secrete a sufficient level of SASP factors to surrounding cells because they are immediately eliminated from the epithelia. In contrast, additional TP53 mutation allows the mutant cells to remain on the epithelia, allowing for continuous secretion of SASP factors to neighbouring cells. We have discussed these points in the revised manuscript.

What happens to cdkn2a/b after complete abrogation of p53 in RasG12V expressing cells?

Response: Based on this question, we examined the expression levels of *cdkn2a/b* in TP53-lacking Ras^{G12V} cells. As shown in revised Supplementary Fig 6c, deletion of TP53 protein expression by injection of *tp53* MO did not affect *cdkn2a/b* expression levels in Ras^{G12V} cells, suggesting that induction of “strong” senescence in Ras^{G12V}-TP53^{R175H} double-mutant cells was not due to TP53 loss-of-function.

Il1b and ROS are pointed out as responsible for tumour formation since MO against il1b and ROS scavenger NAC limit tumour mass formation. Could the authors recapitulate tumour mass production by RasG12V expression combined with il1b or increased ROS?

Response: We thank Reviewer#1 for this thoughtful comment. As suggested, we examined whether tumour cell mass formation is recapitulated by Ras^{G12V} expression combined with IL-1 β and/or ROS. We treated the larvae with hydrogen peroxide (H₂O₂), an ROS which can be generated in senescent cells (Gorgoulis et al. *Cell* 2019). As shown in revised Fig 5d, treatment with H₂O₂ alone did not facilitate cell mass formation. In contrast, co-expression of IL-1 β in Ras^{G12V} cells induced cell mass formation. Furthermore, H₂O₂ treatment enhanced this cell mass formation in larvae with IL-1 β -overexpressing Ras^{G12V} cells. These results indicate that tumour cell mass formation is recapitulated by Ras^{G12V} expression combined with IL-1 β and ROS, and that IL-1 β -mediated cell proliferation and ROS-mediated secondary senescence synergistically promote cell mass formation.

One possible alternative explanation for the observations is that RasG12V-expressing cells are unstable and are eliminated, while p53 mutant provides a survival activity to RasG12V cells, allowing them to produce and secrete these factors for a longer period of time, leading to tumour formation.

Response: As described above, we agree with Reviewer#1's comments and have discussed these points in the revised manuscript.

Have the authors checked apoptosis in their systems?

Response: As shown in revised Supplementary Fig 1b, one of the major apoptosis markers, active caspase 3, was not detected in mosaically introduced Ras^{G12V} cells, suggesting that Ras^{G12V} cells are eliminated in an apoptosis-independent manner.

Then the authors start a series of experiments and claims that to this reviewer are confusing. On the one hand, they claim that RasG12V-p53R175H cells induce features of senescence in neighbouring cells, such as ROS production, but at the same time they claim they induce proliferation, as measured by phosphor-histone H3. Could the authors further clarify this issue?

Response: We apologize for the confusion. Although the double-mutant cells secreted IL-1 β and ROS, which can induce proliferation and senescence, respectively, in neighbouring cells, we considered that these processes would not occur simultaneously in the same neighbouring cells. To confirm this prediction, we investigated the expression of a cell proliferation makers (phospho-histone H3 (pH3) and EdU incorporation) and senescence markers (γ H2AX and *cdkn2a/b*) in the neighbours of Ras^{G12V}-TP53^{R175H} double-mutant cells. As shown in revised Supplementary Fig 7b and 9f, both senescence marker-positive cells and proliferation maker-positive cells were present in the Ras^{G12V}-TP53^{R175H} cell neighbours, and all senescence maker-positive cells neighbouring Ras^{G12V}-TP53^{R175H} cells were proliferation marker-negative, suggesting that cellular senescence and cell proliferation occur in different cells.

Similarly, they claim that the tumour masses contain cells that show markers of what they consider senescence (although this claim is not supported by proliferation markers, for example) and at the same proliferate to form the tumour. And they might carry RasG12V-p53R175H or not.

Response: Thank you for these helpful comments. As shown in revised Supplementary Fig 7a, we confirmed that EdU-positive proliferating cells were present inside the Ras^{G12V}-TP53^{R175H}-induced cell mass and that these cells did not carry Ras^{G12V}-TP53^{R175H}.

Double mutant RasG12V-p53R175H do not produce tumour masses in p53KD ZF skin and they claim that this means that induction of paracrine senescence is required for tumour formation. Have the authors checked what happens when they introduce RasG12V alone in p53KD skin?

Response: Indeed, we checked this point. As shown in original and revised Fig 3d, mosaic introduction of cells with Ras^{G12V} alone hardly produced a tumour-like cell mass in *tp53*-knockdown (*tp53* MO-injected) skin.

Authors also show that ROS and il1b are uniformly expressed throughout the tumour masses and claim that this means that senescence is propagated by senescent neighbouring cells, which is a speculative claim difficult to reconcile with the proliferation required to form a tumour.

Response: Reviewer#1 pointed out that ROS and *il1b* are uniformly expressed throughout the tumour-like cell mass, which contains senescent cells (but not proliferative cells). However, as shown in revised Fig 5e and i (corresponding to original Supplementary Fig 3a and c), their expression was not uniform. The cell mass contained cells lacking expression of ROS and *il1b*, indicating that non-senescent cells were present in the cell mass. In addition, the double-mutant cells induced EdU-or pH3-positive proliferating cells in the surrounding area (revised Fig 2c, 5a) and EdU-positive cells were detected in the cell mass (revised Supplementary Fig 7a). All pH3-positive proliferative double-mutant cells were γ H2AX-negative non-senescent cells (revised Supplementary Fig 7b), suggesting that proliferation and senescence occur in a mutually exclusive manner.

In addition, we performed further analyses to confirm the propagation of senescence. As shown in revised Fig 2b, 5k, and Supplementary Fig 10d, introduction of Ras^{G12V}-TP53^{R175H} double-mutant cells, but not Ras^{G12V} single-mutant cells, strongly induced SA- β -gal-positive senescent cells in the area adjacent to the double mutant cells (within a 20- μ m radius from double-mutant cells). Introduction of double-mutant cells also moderately induced SA- β -gal expression in the area within a 20–40- μ m radius from the double-mutant cells, but not in the area outside of a radius of 40 μ m from these cells. We also successfully generated *cdkn2a/b*-reporter zebrafish, in which *cdkn2a/b* (senescent marker gene) expression was converted to fluorescence by the fast-maturing fluorescent protein “Achilles”, which revealed that the neighbouring cells gradually showed upregulated expression of *cdkn2a/b* (revised Supplementary Fig 9e, f and Supplementary Movie 5) Notably, to our knowledge, this is the first report using fluorescent-based live imaging to show that normal cells become senescent cells in living tissue. These results support that the double-mutant cells propagate senescence to their neighbours.

Introduction of RasG12V expressing cells in mutant p53R175H skin leads to the formation of tumour masses. This would imply that now RasG12V expression alone does not lead to the elimination of these cells and results in “strong” senescence. Since mutant p53 is not present in the same cell than mutant Ras, this implies that the expression of some factor/s is affecting Ras expressing cells. Have the authors identified this factor/s?

Response: We apologize for the confusion. In this experimental system, “Ras^{G12V} mutation” is mosaically introduced into the TP53^{R175H} skin. Therefore, Ras^{G12V}-introduced cells also express TP53^{R175H}. To clarify this point, we have modified the description of these experiments in the revised manuscript.

*Have they tried to decrease *il1b* in *p53R175H* mutant cells or block ROS production?*

Response: Based on Reviewer #1 suggestion, we tested whether *il1b* knockdown (*il1b* MO) or blocking of ROS production (NAC treatment) inhibits cell mass formation in TP53^{R175H}-expressing skin with mosaically introduced Ras^{G12V}. As shown in revised Fig 6b and 6c, these treatments decreased the number of larvae with cell masses, suggesting that IL-1 β and ROS contribute to cell mass formation in this context.

*Finally, the authors use damaged epithelia (doxorubicin treated ZF) to claim that a senescent tissue does not retain the ability to eliminate RasG12V expressing cells and this leads to tumour formation. Do RasG12V expressing cells show increased *cdkn2a/b*, *il1b*, *il11b* or ROS?*

Response: To answer this question, we investigated whether doxorubicin (Doxo) treatment increases the levels of *cdkn2a/b*, *il1b*, *il11b*, or ROS. As shown in revised Fig 6d–g and Supplementary Fig 11f, Doxo treatment enhanced the expression of *cdkn2a/b*, *il1b*, and *il11b* and production of ROS in Ras^{G12V} cells, suggesting that, like Ras^{G12V}-TP53^{R175H} cells, the Ras^{G12V} cells on Doxo-treated skin underwent cellular senescence and produced SASP factors.

What is the effect of eliminating p53 in RasG12V expressing cells in the context of doxo-induced senescent skin? Would doxo treatment of p53null skin promote tumour formation by RasG12V expressing cells?

Response: To answer these questions, we examined whether TP53 elimination by *tp53* MO injection affected Ras^{G12V} cell-induced cell mass formation on Doxo-treated skin. As shown in revised Fig 6i, injection of *tp53* MO significantly inhibited cell mass formation in Doxo-treated skin with mosaically introduced-Ras^{G12V} cells,

suggesting that neighbouring cell senescence is required for cell mass formation in Doxo-treated skin.

Reviewer #2 (Remarks to the Author):

We thank Reviewer#2 for the positive comments regarding the interesting account of the response of normal cells to oncogenes using an elegant experimental system. We also appreciate the insightful comments and suggestions provided, particularly those on cellular senescence. These comments are helpful for improving the quality of our paper. As indicated in the responses below, we have considered all comments and suggestions in the revised version of the manuscript.

Major points:

The authors claim that mutant p53 increases the frequency of senescent cells, leading to proliferation of neighboring cells due to SASP. However, not every Ras/p53 mutant cell has senescent markers. Therefore, one interpretation is that mutant p53 is diversifying the population in two ways: (i) increasing proliferation in a non-senescent subpopulation and (ii) increasing hallmarks of senescence in another one. This idea is conceptually reminiscent of a model advanced by Sean Morrison's group (<https://www.nature.com/articles/nature12830>) where they claimed that oncogenic Nras can increase proliferation of a subpopulation of HSPCs, and increase self-renewal of another subpopulation. Other studies have shown that the induction of senescence in one cell type can induce the reprogramming of an adjacent cell (<https://science.sciencemag.org/content/354/6315/aaf4445.long>).

Response: We appreciate these comments indicating the possibility that proliferating Ras^{G12V}-TP53^{R175H} cells exist. To test this possibility, we performed EdU incorporation assays. As shown in revised Fig 2c, more than 30% of control cells were EdU-positive, whereas only 5% of Ras^{G12V}-TP53^{R175H} cells were EdU-positive. In addition, we did not detect any EdU-positive Ras^{G12V}-TP53^{R175H} cells within the cell mass, and most proliferative cells inside the cell mass were cells

lacking Ras^{G12V}-TP53^{R175H} double mutations (Supplementary Fig. 7a). Therefore, we considered that most Ras^{G12V}-TP53^{R175H} cells underwent cell cycle arrest, whereas a few transiently maintained their proliferation activity and then gradually lost this activity.

As Reviewer #2 suggested, introduction of mosaic Ras^{G12V}-TP53^{R175H} cells may also induce reprogramming of neighbouring cells. As shown in Fig. 3c, the Ras^{G12V}-TP53^{R175H} cell-induced cell mass contains a variety of cells with different shapes, sizes, and affinity for haematoxylin and eosin staining, and some cells may have been de-differentiated from neighbouring epithelial cells. In further studies, we would like to examine whether neighbouring cell reprogramming is induced by Ras^{G12V}-TP53^{R175H} cell introduction.

The authors co-introduce the TP53R175H mutation with Ras and suggest this reinforces senescence based on the induction of some damage associated markers, cdkn2a/b, and the expression of inflammatory factors also linked to the SASP. There are a number of problems with this interpretation. For example, Il1b could be induced in any inflammatory setting, and ROS is by no means senescence specific. Certainly while cdkn2a is induced during senescence in mammalian systems, inactivation of p53 leads to aberrant activation of the locus (so here high cdkn2a levels are actually indicative of failure of the program). While having good markers of the senescence program have limited the field, there is so little characterization of the cells in the zebrafish model that the authors' bold claims relating to the biology are preliminary.

Response: We appreciate Reviewer #2's helpful comments. To assess cellular senescence and SASP in the zebrafish model, we additionally evaluated well-known senescence markers (SA-β-gal, reduction of EdU incorporation, and trimethylation of histone 3 at lysine 9 (H3K9me3)) and SASP factors (IL-6 and CXCL8/IL-8) in mammalian systems certified by the International Cell Senescence Association (Gorgoulis et al. *Cell* 2019). We detected SA-β-gal activity (revised Fig 2b), reduced EdU incorporation (revised Fig 2c), H3K9me3 (revised Supplementary Fig 2a), and IL-6 (*il6*) and CXCL8 (*cxc18a*) expression (revised Supplementary Fig 2b) in Ras^{G12V} cells in zebrafish larval skin, and the additional TP53^{R175H} mutation enhanced all of these factors. These results strongly support our hypothesis that cellular senescence and SASP factor production occurs in Ras^{G12V} cells and Ras^{G12V}-TP53^{R175H} double-mutant cells.

Related to the above point, there is a paucity in data regarding cell proliferation in oncogene expressing cells, despite the fact that this is one of the classical phenotypic outcomes attributed to both oncogenes and senescence. Are oncogene-expressing cells dividing or not?

Response: As described above, we carefully checked the cell division activities of oncogene-expressing cells by performing an EdU incorporation assay. As shown in revised Fig 2c, around 30% of mCherry-expressing cells (control cells) and TP53^{R175H} cells were EdU-positive, whereas 5.7% and 5.2% were EdU-positive Ras^{G12V} cells and Ras^{G12V}-TP53^{R175H} cells, respectively, indicating that cell division was stopped in almost all of these cells.

Does cdkn2a/b activation correlate with decreased division (in the background of oncogenic Ras, or Ras+TP53)?

Response: Indeed, we confirmed that *cdkn2a/b* expression correlates with decreased division in a newly developed *cdkn2a/b* reporter zebrafish line useful for visualizing endogenous expression of *cdkn2a/b* as fluorescence of a faster-maturing YFP variant Achilles. As shown in revised Supplementary Fig 9c, both *cdkn2a/b* reporter-positive Ras^{G12V} single-mutant and Ras^{G12V}-TP53^{R175H} double-mutant cells were completely EdU-negative. Consistent with these observations, live-imaging analysis showed that *cdkn2a/b* reporter-positive Ras^{G12V}-TP53^{R175H} cells did not divide from day -1 to day -2 (data not shown).

Are dividing or arrested cells the ones that are producing SASP?

Response: To answer this question, we investigated the correlation between cell division and SASP factor production. We focused on IL-1 β as a SASP factor because it is one of the major SASP factors and plays an important role in tumour-like cell mass formation in this context. We performed an EdU incorporation assay in an IL-1 β reporter zebrafish line, Tg(*il1b*:EGFP), which express GFP under control of the endogenous IL-1 β (*il1b*) gene promoter (Hasegawa et al., *Elife* 2017). As shown in revised Supplementary Fig 2d, more than 40% of Ras^{G12V}-TP53^{R175H} double-mutant cells highly expressed the IL-1 β reporter and all IL-1 β reporter-positive double-mutant cells were EdU-negative, suggesting that IL-1 β -

producing double-mutant cells are arrested (senesced). Therefore, we considered that the senesced double-mutant cells produced IL-1 β as a SASP factor, thereby stimulating formation of tumour-like cell masses.

Figures 3D and 5F present very similar experiments, but one is used to imply that p53 knockdown in oncogene expressing cells is insufficient to generate a tumor-like mass, whereas the other one is used to conclude that p53-dependent secondary senescence is important for tumor growth. Perturbation of p53 specifically in oncogene-expressing cells would be necessary to disentangle these two experiments and conclusions.

Response: We thank Reviewer#2 for this insightful comment. To resolve this concern, we developed methods for blocking TP53 activity in oncogene-expressing cells using the TP53 deletion mutant (TP53^{DD}) which can form an oligomer with endogenous TP53 proteins and function in a dominant-negative manner (Bowman et al., *Genes Dev* 1996). We first confirmed that TP53^{DD} could perturb the activity of wild-type TP53 in a dose-dependent manner in culture cells (revised Supplementary Fig 6b). Next, we co-expressed TP53^{DD} in Ras^{G12V} cells to inhibit endogenous TP53 activity. As shown in revised Fig 3d, co-expression of TP53^{R175H} significantly enhanced Ras^{G12V} cell-induced cell mass formation, whereas co-expression of TP53^{DD} did not affect this expression. These results reinforce that gain-of-function but not loss-of-function of TP53 cooperates with the Ras^{G12V} mutation to promote tumour-like cell mass formation in our experimental system.

The mechanism of oncogene expressing cell elimination is not clear though as the authors suggest it does not appear to be immune mediated. The authors further state that their data “indicates that cellular senescence can suppress tumorigenesis through immune cell-independent oncogenic cell elimination”. All of the data here are correlative and there is no direct evidence for this statement.

Response: In zebrafish, the adaptive immune system become functionally mature at approximately 4 weeks post-fertilization (Trede et al., *Immunity* 2004). Zebrafish larvae at 1 dpf, in which Ras^{G12V} cell elimination was observed in this study, had not developed an adaptive immune system at this time point. Although

1 dpf larvae contained developing macrophages, monocytes, and neutrophils, we found that *spi1b* MO injection almost completely depleted these immune cells (revised Supplementary Fig 1c) but did not affect Ras^{G12V} cell elimination (revised Supplementary Fig 1d, corresponding to original Supplementary Fig 1b). Therefore, we concluded that Ras^{G12V} cell elimination in zebrafish larva occurs independently of the immune system.

In this study, we showed that communication with neighbouring normal cells drives Ras^{G12V} cell elimination by inducing cellular senescence. In the original manuscript, we showed that mosaic Ras^{G12V} cells undergo senescence and consequent cell swelling and adhesion loss in a neighbouring normal cell-dependent manner. In the revised manuscript, we also describe the machinery involved in neighbouring cell-dependent Ras^{G12V} cell senescence. Recent studies by Yasuyuki Fujita's group reported that mosaically introduced Ras^{G12V} cells are eliminated from the mammalian epithelia through calcium channel (mechanosensitive calcium channel and IP3 receptor)-dependent Ras^{G12V} cell-neighbouring cell communication and neighbouring cell-mediated activation of PDK (pyruvate dehydrogenase kinase) in Ras^{G12V} cells, and that pharmacological inhibition of calcium channel or PDK blocks the Ras^{G12V} cell elimination (Takeuchi et al., *Curr Biol* 2020; Kon et al., *Nat Cell Biol* 2017). As shown in revised Supplementary Fig 12a and b, pharmacological inhibition of the calcium channel or PDK blocked Ras^{G12V} cell senescence and elimination, suggesting that the calcium channel and PDK work upstream of Ras^{G12V} cell senescence, and the mechanism of zebrafish senescence-driven Ras^{G12V} cell elimination may be similar to that of mammalian Ras^{G12V} cell elimination.

The data interpretation of the results that p53R175H acts through a gain of function are not decisive and difficult to interpret. As I understand the experiment the results in otherwise p53 wt background, so p53 could simply be activating as a dominant negative. While apparent knockdown of wt p53 in that system does not enhance tumor growth, but we don't know whether there could be simply quantified differences on p53 output. More decisive comparisons (and evaluation of output) would be needed to relate this to p53' gain of function'. Of course then one would wonder how that actually worked.

Response: To evaluate whether TP53^{R175H} acts through a gain-of-function, we

compared the effects of exogenous TP53^{R175H} on Ras^{G12V} cells with those of exogenous TP53 wild-type (TP53^{WT}) or TP53 dominant-negative mutant (TP53^{DD}) cells. As shown revised Supplementary Fig 6d, co-expression of TP53^{R175H} enhanced Ras^{G12V}-induced expression of *il1b* and *il1b*, whereas co-expression of TP53^{WT} or TP53^{DD} did not affect this expression. Consistent with these results, cells expressing both exogenous Ras^{G12V} and TP53^{R175H} efficiently induced tumour-like cell mass formation compared to cells expressing exogenous Ras^{G12V} and TP53^{WT} (revised Fig 3e). These results suggest that TP53^{R175H} exerted a gain-of-function rather than a loss-of-function in Ras^{G12V} cells in zebrafish skin.

Furthermore, we confirmed that co-expression of other well-known TP53 gain-of-function mutants, TP53^{R273H} and TP53^{R248W}, also blocked elimination of Ras^{G12V} cells and enhanced Ras^{G12V} cell-induced tumour-like cell mass formation, as observed for TP53^{R175H} (revised Supplementary Fig 5b, c). These results are consistent with our model showing that additional TP53 gain-of-function mutation changes the Ras^{G12V} cell behaviour.

Reviewer #3 (Remarks to the Author):

We thank Reviewer #3 for the positive comments on our findings, and for giving us important and concrete suggestions, which were helpful for improving our manuscript. As indicated in the responses below, we have carefully considered all comments and suggestions in the revised version of the manuscript.

Major Points

1) *It seems that all of the oncogenic cells are present in the head region of the zebrafish larvae. Is this the case? It would be helpful to clarify if there is something specific about this area, or did the authors just choose to focus on this region for analysis. Likewise, it is not clear where on the animal that the cell masses formed. Presumably, it is in the same area? This should be mentioned in the text, and potentially illustrated with a schematic.*

Response: Oncogenic cells were introduced not only to the head region but also to the trunk and tail region. Oncogenic elimination and cell mass formation also

occur in all areas. We described information on the region in the revised figure legends.

2) *For several quantifications, the legend states that each dot represents an individual embryo. Yet, the number of embryos analyzed seems low given the high-throughput nature of the system (Fig 1 B, 2D, 3A). Likewise, it is not clear if these embryos came from three independent experiments. Additional analysis should be performed to bolster these claims and information should be added to the figure legends to clarify if embryos from independent experiments were analyzed.*

Response: As Reviewer #3 suggested, we have added the larvae analysed in revised Fig 1b, 2e, and 3a (corresponding to original Fig 1b, 2d, and 3a, respectively). We also clarified the descriptions of the multiple independent experiments in revised Supplementary table 2.

3) *The authors state that “The oncogenic cells appears to occur in an immune cell-independent manner”. While the authors use spi1b to suppress myeloid lineage formation, no controls are provided to show this perturbation was effective. It would be important to show macrophages and/or neutrophils are normally present at this time in development and are significantly depleted by the spi1b MO to support their claims.*

Response: As Reviewer #3 suggested, we confirmed that immune cells are present in normal zebrafish larvae and *spi1b* MO can block their development. As shown in revised Supplementary Fig 1c, expression of the macrophage/monocyte marker *lcp1* (Berman et al. *Exp Hematol* 2005) and neutrophil marker *npsn* (Di et al. *Open Biol* 2017) were detected in 1 dpf larvae, in which Ras^{G12V} cell elimination occurred, whereas *spi1b* MO injection almost completely eliminated their expression. These results suggest that macrophages, monocytes, and neutrophils are generated in normal 1 dpf larvae and that *spi1b* MO can block their generation. These results also support strengthen our hypothesis that Ras^{G12V} cell elimination is immune cell-independent in this context.

4) *This also brings up the interesting question of how the oncogenic cells are*

eliminated. The authors cite a recent paper from Takeuchi et al that provides mechanistic insight into how epithelial cell extrusion is regulated to clear unfit cells. Can the authors comment if that is the case here?

Response: As suggested, we investigated whether the previously reported mechanisms of epithelial cell extrusion regulate senescence-driven Ras^{G12V} cell elimination, which was observed in this study. We found that the previously identified regulators, calcium channel and PDK (pyruvate dehydrogenase kinase) (Takeuchi et al., *Curr Biol* 2020; Kon et al., *Nat Cell Biol* 2017), are involved in senescence-driven Ras^{G12V} cell elimination. As shown in revised Fig 12a and b, pharmacological inhibition of the calcium channel or PDK blocked nonautonomous Ras^{G12V} cell senescence and elimination, suggesting that the mechanism reported previously by Takeuchi et al. and Kon et al. is similar to that of senescence-driven Ras^{G12V} cell elimination. We have discussed these points in the revised manuscript.

Along these lines, Figure 1c nicely shows rounded KRAS+ cells, but it would be useful to show neighboring healthy GFP+ cells as well to fully demonstrate that the KRAS cell is out of the plane of tissue, and if it is actively being eliminated by the epithelial neighbors.

Response: As Reviewer #3 suggested, we showed the apically protruding Ras^{G12V} cells and neighbouring cells in revised Fig 1c. Additionally, investigation of the action of neighbouring cells showed that these cells accumulated F-actin along the contact site of Ras^{G12V} cells, whereas the neighbours of Ras^{G12V}-TP53^{R175H} double-mutant cells did not induce F-actin accumulation (revised Supplementary Fig 1e). These observations suggest that the neighbouring cells play an active role in Ras^{G12V} cell elimination.

5) Are the KRAS+ cells undergoing apoptosis (i.e., positive for cleaved caspase 3 or Tunel) during elimination from the tissue? This could provide clues as to the mechanisms used to eliminate these cells from the tissue and may also support the idea that the additional TP53 mutations suppress cell death and drive senescence.

Response: As Reviewer #3 suggested, we investigated whether apoptosis

occurred in Ras^{G12V} cells. As shown in revised Supplementary Fig 1b, Ras^{G12V} cells including those protruded from the epithelia were cleaved caspase 3-negative. This result suggests that apoptosis is not involved in eliminating Ras^{G12V} cells.

6) The authors frequently refer to KRAS+ TP53 R175H double mutant cells. While KRAS is fluorescently tagged, overlap with a fluorescently tagged TP53 R175H is never shown. Given the mosaic nature of injections, expression of these two constructs in the same cell may not always be the case. Alternatively, it could be that activated TP53 in healthy neighbors influences the KRAS cells, a point addressed later in the paper. It would be helpful for the authors to demonstrate an increase in TP53 R175H levels and determine what cell type is it expressed in.

Response: Thank you for this thoughtful comment. We agree that it is important to consider the mosaic nature of the injection system. However, we considered that most of the co-injected TP53^{R175H} construct was expressed in Ras^{G12V} cells but not in neighbouring cells. To demonstrate this point, we added data showing the distribution of cells with Ras^{G12V} and TP53^{R175H}, which were labelled with mCherry and GFP, respectively. As shown in revised Supplementary Fig 5a, 93.8% of Ras^{G12V} (mCherry)-positive cells expressed TP53^{R175H} (GFP) in zebrafish larvae injected with the Ras^{G12V} and TP53^{R175H} constructs. In addition, there were no TP53^{R175H}-positive cells neighbouring Ras^{G12V}-positive cells. These results indicate that Ras^{G12V} and TP53^{R175H} were mostly co-expressed in zebrafish larvae co-injected with Ras^{G12V} and TP53^{R175H} constructs in our experimental system.

Additionally, to exclude the possibility that TP53^{R175H} expressed in the neighbour of Ras^{G12V}-expressing cells drove cell mass formation, we tested the effect of a bi-directional promoter construct that can induce the expression of both the Ras^{G12V} and TP53 mutant (TP53^{R175H} or TP53^{QSQS} as a negative control) simultaneously (Fig x1a, see below). As shown in Figure x1a and x1b (see below), cells with the bi-directional constructs expressing both Ras^{G12V} and TP53^{R175H} induced cell mass formation, whereas cells with constructs expressing both Ras^{G12V} and TP53^{QSQS} did not. These results are consistent with our model showing that additional introduction of the TP53^{R175H} mutation into Ras^{G12V} cells, but not their neighbours, induced cell mass formation.

Figure x1. Mosaic introduction of a bi-directional promoter construct driving the expression of both Ras^{G12V} and TP53 mutant can induce cell mass formation.

a Bi-directional UAS vector-mediated co-expression system

b Introduction with Ras^{GV} and TP53^{mut} (by single bi-directional UAS vector)

c Larvae with cell mass (%)

7) On pg 8, the authors state “these results suggest that the addition of the TP53 R175H mutation reinforces RAS G12V induced senescence.” It would be useful to show that TP53 loss of function does not reinforce senescence in this context.

Response: As Reviewer#3 suggested, we confirmed that TP53 loss-of-function does not reinforce Ras^{G12V}-induced senescence. As shown in revised Supplementary Fig 6c, depletion of endogenous TP53 proteins by *tp53* MO injection did not affect the expression levels of the senescence marker *cdkn2a/b* in Ras^{G12V} cells. In addition, we found that co-expression of TP53^{R175H}, but not TP53 wild-type (TP53^{WT}) or TP53 dominant-negative mutant (TP53^{DD}), enhanced Ras^{G12V}-induced expression of the SASP factors, *il1b* and *il11b* (revised Supplementary Fig 6d). These results support our conclusion that addition of the TP53^{R175H} mutation reinforces RAS^{G12V}-induced senescence.

8) The authors state that the “effects of the p53 175H mutation on these (ROS, IL1b and IL11b) levels were minor (Figure 2a, 4a, 4b)”. As Figure 2a only shows ROS, to better support this idea, it would be important to show the impact of KRAS alone and TP53 R175H alone on IL1B and IL11B expression specifically within these cells in Figure 4C. Also, is this observed expression increase dependent on ROS levels? Determining if this increase is abolished when suppressing ROS with NAC would help further support this mechanism.

Response: As Reviewer #3 suggested, we added data showing the impact of Ras^{G12V} alone, TP53^{R175H} alone, and Ras^{G12V}-TP53^{R175H} double mutation on *il1b* and *il11b* expression in cells expressing these proteins (revised Fig 4b and Supplementary Fig 2b, c). Introduction of Ras^{G12V} alone slightly activated their expression, whereas the effects of TP53^{R175H} alone were very minor. Importantly, their expression was synergistically enhanced by Ras^{G12V}-TP53^{R175H} double mutation.

According to Reviewer #3's suggestion, we also tested the effects of ROS depletion by NAC treatment on *il1b* and *il11b* expression in the larvae introduced with Ras^{G12V}-TP53^{R175H} double mutation. As shown in revised Supplementary Fig 2e, NAC treatment significantly reduced their expression by 50%. These results are consistent with our model showing that double-mutant cells produced ROS to mediate secondary senescence and SASP factor (*il1b* and *il11b*) expression in neighbouring cells.

9) For IL1B MO experiments in Figure 5C, an important control would be to show IL1b (but not other SASPs) is no longer induced or that expression is significantly diminished.

Response: We appreciate this important suggestion. In this study, we used *il1b* translation-blocking MO, which depletes endogenous IL1b proteins but not the transcripts. Because there are no commercially available antibodies that can detect zebrafish IL1b proteins efficiently, we applied a conventional approach for MO validation by evaluating the expression levels of the target gene fused in-frame with the fluorescent protein gene. As shown in revised Supplementary Fig S8a, mRNA containing the 5' UTR and 5' coding region of *il1b* fused in-frame with the GFP gene was injected into zebrafish embryos with control MO or *il1b*

MO. Control MO-injected embryos showed GFP fluorescence, whereas *il1b* MO injection significantly diminished GFP fluorescence, suggesting that *il1b* MO suppressed endogenous *il1b* expression.

In contrast, we could not confirm that *il1b* MO does not affect the protein levels of other SASP factors because there are no antibodies for detecting other zebrafish SASP factors, such as IL-11b and IL-6. However, we confirmed that, among zebrafish transcripts, only *il1b* transcripts possess the translation initiation region which can be efficiently hybridized with *il1b* MO according to a BLAST search. Therefore, *il1b* MO appears to specifically block the translation of *il1b* but not that of other genes.

10) The authors nicely show the formation of cell masses, but it is not clear whether changes in cell-cell adhesion occur within the cell mass. What is the status of E-cadherin levels in KRAS+ TP53 R175H positive cells and their neighbors within the cell mass?

Our Response:

In response to Reviewer #3's question, we examined the status of E-cadherin levels in Ras^{G12V}-TP53^{R175H} double-mutant cells and those within the cell mass. As shown in revised Supplementary Fig 4c, membrane E-cadherin levels in Ras^{G12V}-TP53^{R175H} double-mutant cells were higher than those in Ras^{G12V} single-mutant cells. Interestingly, membrane E-cadherin levels in cells within the cell mass were heterogenous (revised Supplementary Fig 4d). Some double-mutant cells and their neighbours showed high levels of E-cadherin, whereas other cells expressed low levels of E-cadherin. These results indicate that additional TP53^{R175H} mutation reversed Ras^{G12V}-induced E-cadherin reduction, but some Ras^{G12V}-TP53^{R175H} cells and neighbouring cells exhibited reduced E-cadherin levels after forming a cell mass. Previous studies showed that a SASP factor, including ROS, can disrupt the membrane-localization of cadherin (van Wetering et al., *J Cell Sci* 2002 115 (9): 1837–1846.). ROS, which are continuously produced in the cell mass, may have reduced E-cadherin levels in some cells in the mass. We have discussed this point in the revised manuscript.

11) The authors argue that the double mutant KRAS p53 R175H mutant cells induce senescence in the neighboring cells, supported by IL1b and ROS being broadly expressed in the cell mass (Supp Fig 3a,c). Based on this result, then

one would predict that induced ROS and IL1B expression in the neighboring cells would be suppressed in the TP53 MO skin. Is this indeed the case?

Response: Thank you for this thoughtful comment. Based on this question, we performed additional experiments. As shown in revised Supplementary Fig 10b and c, we confirmed that *tp53* MO suppressed ROS production and *il1b* expression in the neighbouring cells but not in Ras^{G12V}-TP53^{R175H} cells.

12) The authors use doxorubicin to create senesced epithelia, and while quite intriguing, this new model is not fully characterized making it difficult to interpret the findings. In line with the rest of the manuscript, it would be important to show markers of senescence (ie., Ros dye and IL1b levels) are increased after Dox exposure. Also, does this treatment induce apoptosis in the epithelial cells with DNA damage, as this could confound the interpretation of the results.

Response: As Reviewer #3 suggested, we analysed the expression of senescence and apoptosis markers in Doxo-treated larvae. As shown in revised Fig 6d–g and Supplementary Fig 11d–f, Doxo treatment increased senescence marker *cdkn2a/b*-positive cells and enhanced the expression of SASP-related genes, *il1b*, and *il11b*, and ROS production, but did not induce active caspase-positive apoptotic cells. We also found that *tp53* MO, which can inhibit senescence in neighbouring cells, suppressed cell mass formation in Doxo-treated larvae introduced with Ras^{G12V} cells (revised Fig 6i). These results suggest that cell mass formation by Doxo-treatment and Ras^{G12V} cell introduction shares the same mechanism with the Ras^{G12V}-TP53^{R175H} cell-induced cell mass formation.

Minor points:

It would be helpful to use the TP53 R175H notation in the figures, as opposed to TP53+, which could be interpreted as containing wild type TP53

Response: We improved the notation of TP53 wild-type and mutants in the figures (e.g., TP53^{RH+} and TP53^{WT+}).

For consistency and to aid colorblind readers, the red images in Figure 4c should be made magenta

Response: We converted the red images in original Fig 4c to magenta images in revised Fig 4b and Supplementary Fig 2b.

The data in Supp Fig 3 supports the idea that TP53 R175H promotes accumulation of senescent cells to form masses. Given the mechanistic link with ROS and Il1b later in the manuscript, including this data in the main figures (especially a-c) would help to focus the message for the reader.

Response: Based on Reviewer #3's suggestion, we moved original supplementary Fig 3a–c to the main figure (revised Fig 5e, g, i).

On pg9, the authors state “Surviving double mutant zebrafish skin secreted SASP factors”. While the authors nicely show induced expression of SASP factors, the data presented does not demonstrate that these factors are indeed secreted. The authors should change the language to reflect this fact.

Response: Following Reviewer #3's suggestion, we revised this sentence as follows: “Surviving double-mutant cells in zebrafish skin express SASP factors.”

REVIEWERS' COMMENTS

Reviewer #1 (Remarks to the Author):

This is a striking manuscript with intriguing results. I am not completely convinced about the author's interpretation of some of the experiments but I think the revised version is more complete and addresses a big proportion of the concerns raised by the reviewers. Also, I think it merits publication to allow the scientific community to corroborate these results and further explore their implications.

Reviewer #3 (Remarks to the Author):

The authors have adequately addressed all of my previous comments from the original submission. The new data and additional clarifications have significantly strengthened the manuscript. I recommend publication of the manuscript and congratulate the authors on this intriguing study.

Reviewer #4 (Replacement Reviewer for Reviewer #2, Remarks to the Author):

This is a very interesting study. The authors successfully addressed the comments of Reviewer 1.